# The dominance of large-scale phase dynamics in human cortex, from delta to gamma

David M Alexander[1,2]*, Laura Dugué[1,3]

[1]Université Paris Cité, CNRS, Integrative Neuroscience and Cognition Center, Paris, France; [2]GHU-Paris Psychiatrie et Neurosciences, Hôpital Sainte-Anne, Paris, France; [3]Institut Universitaire de France (IUF), Paris, France

## eLife Assessment

This study introduces a novel method for estimating spatial spectra from irregularly sampled intracranial EEG data, revealing cortical activity across all spatial frequencies, which supports the global and integrated nature of cortical dynamics. It showcases **important** technical innovations and rigorous analyses, including tests to rule out potential confounds. However, further direct evaluation of the model, for example by using simulated cortical activity with a known spatial spectrum (e.g., an iEEG volume-conductor model that describes the mapping from cortical current source density to iEEG signals, and that incorporates the reference electrodes and the particular montage used), would even further strengthen the **solid** evidence.

**Abstract** The organization of the phase of electrical activity in the cortex is critical to inter-site communication, but the balance of this communication across large-scale (>8 cm), macroscopic (>1 cm), and mesoscopic (1 cm to 1 mm) ranges is an open question. The spatial frequencies (i.e. the spatial scales) of cortical waves have been characterized in the gray matter for micro- and mesoscopic scales of cortex and show decreasing spatial power with increasing spatial frequency. This research, however, has been limited by the size of the measurement array, thus excluding large-scale traveling waves. Obversely, poor spatial resolution of extracranial measurements prevents incontrovertible large-scale estimates of spatial power. We estimate the spatial frequency spectrum of phase dynamics in order to quantify the uncertain large-scale range, utilizing stereotactic electroencephalogram to measure local-field potentials within the gray matter. We take advantage of the large extent of spatial coverage of the cortical sheet, and irregular sampling is offset by use of linear algebra techniques. We find the spatial power of the phase is highest at the lowest spatial frequencies (longest wavelengths), consistent with the power spectra ranges for micro- and mesoscale dynamics, but here shown up to the size of the measurement array (up to 8–16 cm). This result arises across a wide range of temporal frequencies, from the delta band (1–3 Hz) through to the high gamma range (60–100 Hz).

*For correspondence:
david.murray.alexander@gmail.com

## Introduction

Traveling waves (TWs) have been described in human cortex since at least 1949 (*Goldman et al., 1949*). They are a commonly described class of the more general phenomena of cortical phase dynamics, which we define as the set of all possible phase patterns. TWs may take the form of linear gradients (*Zhang et al., 2018*; *Alamia et al., 2023*; *Alexander et al., 2006*; *Fakche and Dugué, 2024*), phase cones (*Barrie et al., 1996*; *Ramon and Holmes, 2015*), spirals (*Alexander et al., 2019*;

**eLife digest** Brain cells known as neurons communicate via electrical and chemical signals to pass on information within and between neurons. When large groups of neurons fire in a coordinated way, they create brain waves. These waves or oscillations occur over many different-sized regions, from a few neurons to the whole cortex, the large, folded structure that sits over the rest of our brain and fills most of our skulls.

These patterns of dynamic activity help the brain coordinate and integrate information across regions and are involved in cognitive performance, such as motor control, memory and perception. The timing or phase of electrical activity in the brain is important for different areas to communicate effectively. However, it is still unclear if these activity patterns mostly happen at small (1mm-1cm), medium (>1cm) or large distances spanning almost the entire cortex (>8cm).

While most studies have focused on small and medium waves, which are implicated in sleep, attention and memory, large waves have been understudied. This is mainly due to technical limitations of sampling large areas of cortex and the blurring that occurs when measurements are made from outside the skull.

Alexander and Dugué used existing brain data from epilepsy patients, recorded with a technique called stereotactic electroencephalogram (sEEG). Local brain activity from a large number of electrodes was measured while participants engaged in a delayed free-recall task. The researchers then applied a new method to estimate the spatial frequency spectrum to quantify waves over all scales, from small to large.

Alexander and Dugué discovered that large ripples covering most of the cortex are the strongest and dominate how electrical activity is structured. As a result, signals recorded from a single point in the brain mostly reflect engagement in global brain activity rather than just nearby neurons. This pattern is consistent across many brain frequencies, from slow to fast oscillations. Since brain activity is expensive in terms of metabolism, the existence of these large ripples could be functionally important.

The study of Alexander and Dugué provides new insight into how the brain coordinates communication across different regions. A deeper understanding of the most important factors involved in neuronal processing will contribute to the development of new, targeted medical treatments and technologies to support people living with a range of cognitive conditions.

---

*Prechtl et al., 2000*; *Bhattacharya et al., 2022*), and may also vary in velocity, that is ranging between standing and pure traveling waves (*Alexander et al., 2016*; *Grabot et al., 2022*). Large-scale TWs, specifically, are monotonic phase gradients measured over more than 8 cm, up to the global scale of the cortical sheet.

Theoretical analyses have long predicted the existence of macroscopic wave dynamics (*Nunez, 1974*; *Wright et al., 2001*; *Pang et al., 2023*). Technically, the measured waves are group waves in the physical sense (*Born and Wolf, 1980*), that is, we measure the envelope of the heterogeneous activity at some temporal frequency (TF) as it moves through the cortex. The waves observed in neural mass models arise through a dynamic balance of excitation and inhibition and activated fields radiate outward from previously activated areas (*Nunez, 1974*; *Wright et al., 2001*; *Nunez and Srinivasan, 2006*; *Srinivasan et al., 1998*; *Liley et al., 1999*). Mathematical analyses indicate activity modes that are associated with damped TWs (*Wright et al., 2001*), and under appropriate boundary conditions, these correspond to large-scale TWs (see *Figure 1*). These waves are best understood as being coordinated by long-range myelinated axons (*Nunez and Srinivasan, 2006*). Being group waves, they do not directly indicate the velocity of information transmission (*Nunez, 1974*; *Wright et al., 1994*), that is, the speed of transmission of axonal volleys (c.f. *Srinivasan et al., 1998*; *Liley et al., 1999*). They do, on the other hand, reflect the speed at which regions of activity can flip between different regimes, often as the result of small changes in coordinated synaptic inputs (*Wright et al., 1994*, c.f. *Rosen and Halgren, 2022*).

Alternative hypotheses have been proposed to explain large-scale TWs measured extracranially. We highlight three prior studies (*Orczyk and Kajikawa, 2022*; *Hindriks et al., 2014*; *Zhigalov and Jensen, 2023*) here due to their clear and explicit mathematical treatments, enabling an effective

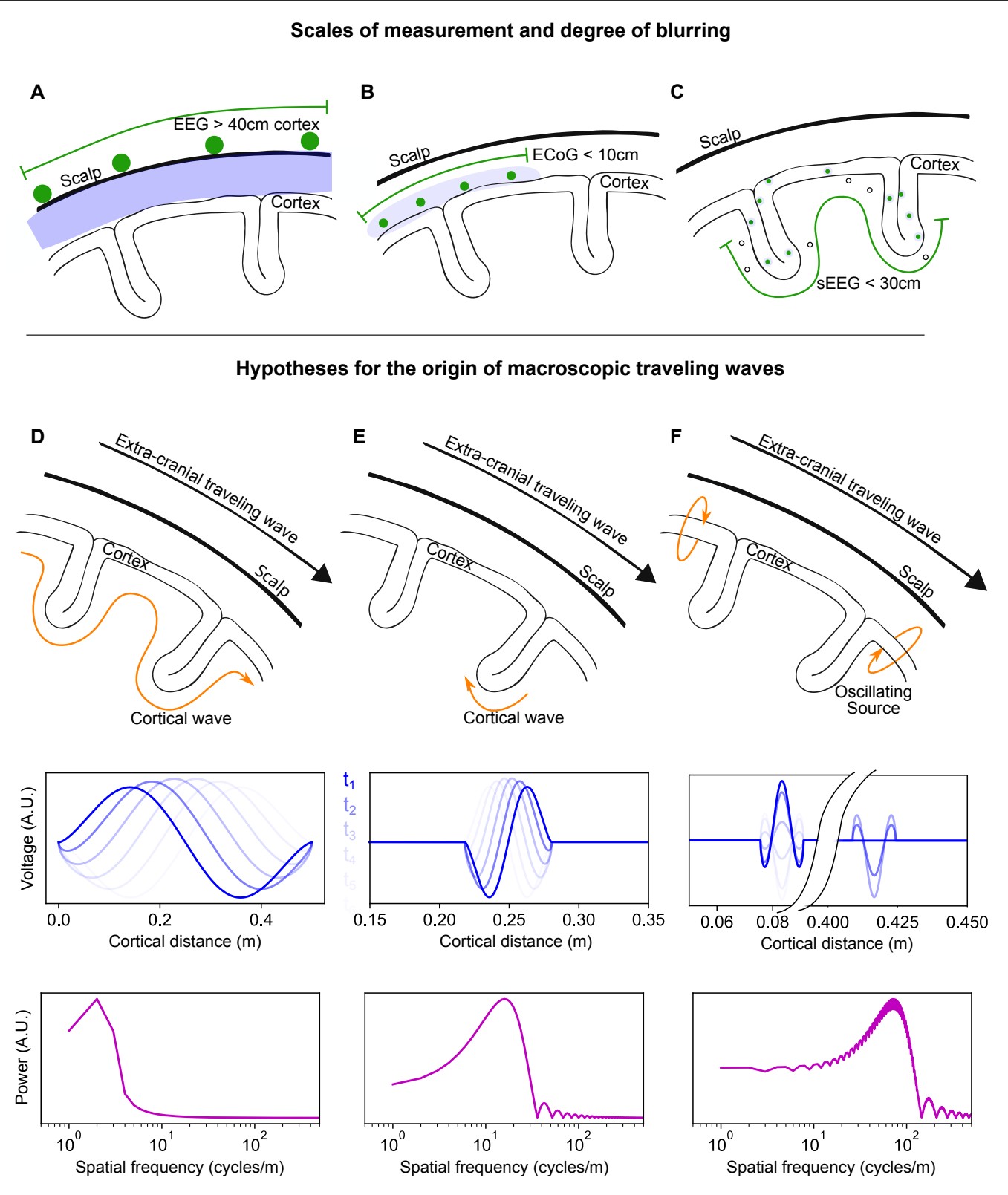

**Figure 1.** Traveling waves, measurements, and origin. (**A–C**) Scales of measurement and degree of blurring in EEG, ECoG, and stereotactic EEG. The maximum linear distances of unfolded cortex typically covered by each technique are shown in each panel. (**A**) EEG measures the electrical activity of the cortex at the surface of the scalp (contacts shown in green). Due to volume conduction in the intervening tissues (blue), the cortical signal is highly blurred and preferentially samples gyri. The EEG array covers the entirety of the scalp in a quasi-regular grid, enabling a large extent of cortical

*Figure 1 continued on next page*

*Figure 1 continued*

coverage (green line, not to scale). (**B**) Electrocorticogram (ECoG) measures the electrical activity of the cortex at the cortical surface. The volume conduction effects are less here, but the measurement array preferentially samples activity at the gyri. The distance between contacts is regular, forming a two-dimensional measurement array, but the size of the array is limited to 10 cm or less.(**C**) Stereotactic EEG measures electrical activity within the gray matter of the cortex, and therefore has a lower degree of blurring. The clinical placement of the linear arrays of contacts results in a highly irregular measurement array, though often with wide coverage of the cortex, though in general less than 30 cm. (**D–F**) Hypotheses for the origin of macroscopic TWs measured extracranially (adapted from *Hindriks et al., 2014*). (**D**) Extracranial TWs reflect real, coordinated, spatio-temporal activity across the entire cortex. The TWs are measurable at the scalp because the dominant SF of the activity has a long wavelength and is therefore detectable at the scalp. A sinusoidal wave corresponding to this hypothesis is shown in blue, with successive time samples in lighter shades. The fast Fourier transform of the model wave is shown in magenta. The expected SF spectrum, when measured in the gray matter, has a peak at a low SF. (**E**) A local wave (shown here in a sulcus) slowly traverses a small region of gray matter via intracortical connections. The wave appears at the scalp as a fast-moving global wave due to volume conduction effects. The wave's (blue) expected SF spectrum (magenta), when measured in the gray matter, has a peak at a high SF. (**F**) Localized oscillatory sources that are out of phase appear as large-scale TWs at the scalp due to volume conduction effects. The oscillators' (blue) expected SF spectrum (magenta), when measured in the gray matter, has a peak at high spatial frequencies corresponding to the size of each oscillatory source, with a weaker elevation in low SF power due to the interaction between sources.

collision with the present research. Local TWs, driven by short-range intracortical connectivity, have been proposed as the origin of the extracranial measurements (*Orczyk and Kajikawa, 2022*; *Hindriks et al., 2014*), as shown in *Figure 1*. Since these waves move slowly in the gray matter, but over a local region, when projected to the scalp they reproduce the fast velocity characteristics of the reported extracranial alpha waves (*Burkitt et al., 2000*; *Ito et al., 2005*; *Klimesch et al., 2007*). Another explanation for macroscopic TWs is in terms of localized oscillating cortical sources mixing at the sensor level (e.g. in EEG). In this approach, two spatially offset and phase-lagged components are derived experimentally by source modeling, which in turn reproduces the measured extracranial waves (*Orczyk and Kajikawa, 2022*; *Zhigalov and Jensen, 2023*; see *Figure 1*). Importantly for the present research, the dynamics associated with the alternative hypotheses can be re-represented as Fourier spectra of gray matter activity, and these spectra give distinct and measurable predictions.

For the veridical TWs hypothesis, the expected peak in the spatial frequency (SF) spectrum of phase is at a low SF, while the alternative hypotheses for artefactual large-scale TWs predict peaks at higher SFs. In order to test these hypotheses, we therefore estimate the relative power of different bands of SFs comparing high ($SF > 16$ cycles/m) to low ($8 < SF < 16$ cycles/m) SF contributions to the measurable phase; these SFs are approximately equivalent to wavelengths ($\lambda = 1/SF$) of $\lambda < 6$ cm vs. $6 < \lambda < 12$ cm.

The alternative explanations that have been proposed for measurements of extracranial macroscopic TWs rely on two assumptions. First, for the localized activity to meaningfully reproduce the sensor level measurements, the activity proposed must dominate during the event; that is, the local TWs or oscillatory sources must arise within a relatively quiescent cortex. Otherwise, the local waves or oscillatory sources, as measured extracranially, will be washed out by the cortical activity that actually dominates. The second, and related assumption, is that measures aggregated over samples and/or trials accurately reflect relevant signals in single sample data (*Alexander et al., 2013*). This aggregation assumption is related to the quiescence assumption because it allows local waves or oscillatory sources to be detected in otherwise 'noisy' environments. Typical sample aggregation procedures, such as averaging over trials in event-related potentials, reverse correlation, or computing the lead-field matrix in source localization, result in activity with high variation across aggregated quantities being made quiescent, thereby treating this activity as noise (*Alexander et al., 2013*). By contrast, the present study estimates SF via low-dimensional representations of the phase matrices using singular value decomposition (SVD), rather than for sample aggregates (e.g. trial means). SVD is a lossless decomposition of a participant's phase data, up to the choice of cut-off rank for exclusion of noise-like phase vectors.

Previous research has consistently revealed that the spatial power of phase decreases monotonically with SF. Intracranial measurements show that the SF spectrum of phase declines with power from 12 cycles/m (8 cm wavelengths; *Zhang et al., 2018*; *Ramon and Holmes, 2015*; *Alexander et al., 2019*; *Alexander et al., 2013*; *Freeman et al., 2003*; *Woolnough et al., 2022*). The estimates, however, use cortical surface arrays with limited spatial coverage (≤8 cm) and were therefore insensitive to longer wavelengths (see *Figure 1*). Likewise, animal studies also point to a SF spectrum for phase that monotonically decreases in power with increasing SF (*Barrie et al., 1996*), but at

measurement scales of 7 mm or smaller (*Barrie et al., 1996*; *Rubino et al., 2006*; *Eckhorn et al., 2004*; *Zanos et al., 2015*). Estimates of the relative contribution of large-scale TWs to the SF spectrum of cortical phase dynamics have used EEG (*Ramon and Holmes, 2015*; *Freeman et al., 2003*; see *Figure 1*) or magnetoencephalogram (MEG; *Alexander et al., 2016*), and consistently describe a SF spectrum with maximum power at wavelengths near the array size, and decreasing in power with smaller wavelengths.

These studies, collectively, have not been able to unambiguously determine the relative importance of low SF components. Fourier analysis is sensitive to the sampling of the underlying data, both at the smallest range (the sampling frequency) and the maximum range (the size of the array; *Cohen, 2014*). Phase dynamics at the largest cortical scales, as broad-band signals, cannot be assessed using small intracortical arrays. Obversely, extracranial measurements are contaminated by volume conduction artifacts (EEG) or blurring by distance to the sensor array (MEG), both of which act as a low-pass filter (*Srinivasan et al., 1998*; *Alexander et al., 2013*; *Orsher et al., 2024*; see *Figure 1*). A low-pass filter will suppress higher SFs, thereby inflating the relative effect of the low SFs quantified from the phase dynamics and introducing an ambiguity. In the present research, we use cortical depth electrodes to assess that part of the SF spectrum of phase between 8 and 50 cycles/m. Stereotactic EEG (sEEG) contacts have excellent spatial resolution and coverage (although sparse and irregular), which can extend to 25 cm of the cortical sheet. By extending gray-matter measurements of the SF spectrum to longer wavelengths, we were able to accurately assess the relative contribution of large-scale phase dynamics to cortical activity.

The numerical methods described in the present research are novel. The spatial pattern of contacts in sEEG within the cortical gray matter has a high degree of irregularity and sparseness. In addition, the contacts are embedded in a thin, three-dimensional sheet. Viewed globally, the sheet covers the surface of a highly convoluted, three-dimensional, distorted spheroid. While methods exist for Fourier analysis on irregular grids (*Margrave and Ferguson, 1997*; *Candes et al., 2008*; *Ying, 2009*; *DSpace, 2025*; *Ruiz-Antolín and Townsend, 2018*), the combination of irregularity, sparseness, and distortion in sEEG makes them unsuitable for the present research question. Methods also exist for computing spatial derivatives on irregular grids, but these generally assume denser sampling within a continuous volume (*Tong et al., 2003*). In the present research, we use a multi-scale, finite difference method to compute the SF of the phase, which is first estimated at each TF. To avoid spurious phase estimates, we first decompose the phase vectors into patterns of spatial covariance using SVD, retaining only spatial components that explain non-trivial amounts of variance.

In the present research, we estimate the SF of the cortical *phase* dynamics. This means removing the amplitude information from the initial Fourier components estimated at each TF. This removal is for two reasons (1): consistency with the research on TWs and other phase dynamics and (2) the assumption that the absolute amplitude measurable at each contact is a source of noise, since it depends on the exact details of contact placement which are not as exact as experimental studies in animals.

Despite the published research on SF spectra of cortical phase dynamics (*Barrie et al., 1996*; *Ramon and Holmes, 2015*; *Freeman et al., 2003*; *Freeman et al., 2000*), decades of neuroscience research has favored explanations of extracranial measurements in terms of mesoscopic TWs or focal cortical sources mixing at the sensor level (*Orczyk and Kajikawa, 2022*; *Hindriks et al., 2014*; *Zhigalov and Jensen, 2023*). Here, we can provide a concrete test of the alternative hypotheses based on a simple criterion. We remove ambiguities in the measurement of purported macroscopic TWs by quantifying them within the gray matter. Consequently, the proposed mechanisms of localized TWs, oscillating sources, or large-scale phase gradients can be tested against the SF spectrum of phase measurements for consistency with the proposed mechanism, as shown in *Figure 1*. Specifically, the characteristically small spatial scale of the proposed mesoscopic waves or oscillating sources will be revealed as maxima (or local maxima) at the higher SFs of the SF spectrum. These high SF components have been hypothesized to combine, via blurring of signal, to form apparent low SF macroscopic TWs at the sensor level (*Orczyk and Kajikawa, 2022*; *Hindriks et al., 2014*; *Zhigalov and Jensen, 2023*). This effect will therefore only be apparent if other cortical activity is relatively quiescent; the high SF components must really be a prominent part of the signal. Alternatively, a finding that low SF components dominate the spectrum of phase, at distances corresponding to large-scale TWs, cannot support the blurring or source mixing explanation for macroscopic TWs. This is because the sensor

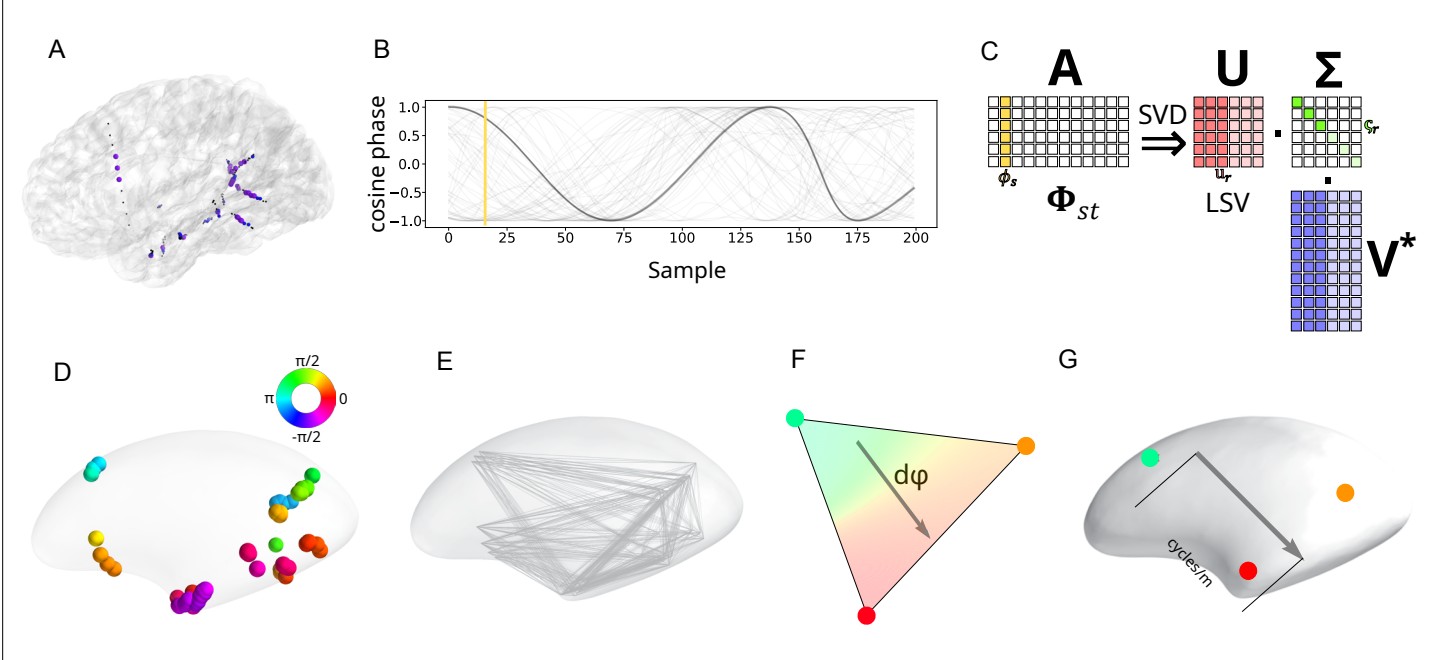

**Figure 2.** Schematic of data processing pipeline. (**A**) Cortical reconstruction of one participant, showing placement of sEEG contacts. Gray matter contacts are shown as black dots. White matter contacts are shown with colored circles (blue, or purple for contacts with 50% lowest amplitude signal). (**B**) The phase values of a segment of data filtered at 9.2 Hz (real part only) are shown for all gray matter contacts in one participant. One contact's phase values are shown in darker gray. A single vector of phase is indicated with the yellow line. (**C**) SVD decomposes any matrix, (**A**), into three separate matrices consisting of a rotation (**U**, the LSVs), a scaling (Σ), and another rotation (**V**, the right singular vectors). Here, the matrix of complex-valued phases for a participant at some center frequency, $\Phi_{st}$, is decomposed into LSVs that represent typical patterns of phase (red vertical bars). The values along the diagonal of Σ give the overall weighting of each LSV in the original matrix. The right singular vectors give the changing time course for each of the LSVs in the original matrix (blue vertical lines). The bright and pale reds, greens, and blues indicate how the least important components (pale) may be dropped, combining a small number of most important components to form a good approximation of the original matrix. (**D**) Placement of gray matter contacts within inflated left hemisphere of cortex. Phase angles of a LSV indicated on the rainbow scale. Phase angles are shown on the color wheel. (**E**) Flattened representation of the approximate equilateral triangles of gray matter contacts, into which the LSV values are resampled as triplets of values. (**F**) Triplets of values from each LSV were used to estimate the rate of change of phase over regions of the cortex. The direction of phase change is indicated with a gray arrow, and phases have been interpolated between vertices of the triangle for visualization purposes. (**G**) The rate of phase change of a triangle of contacts is converted to estimated SF over that region by incorporating the geodesic distances into the numerator. The inflated cortex is shown here, but the geodesic distances are calculated from the folded cortex, that is shown in (**A**).

level measurements will in turn be dominated by the real low SF component of the cortical phase dynamics. Measurement of macroscopic TWs in EEG or MEG can only create a purely artefactual low SF gradient of phase, through mixing and blurring, if no such gradient dominates the underlying signal in the first place (compare *Figure 1A and D*).

## Results

### Overview

In the present research, we analyze a publicly available data set (RAM Public Data https://memory.psych.upenn.edu/RAM_Public_Data). We report the results from 23 participants. Note that most subsequent procedures (construction of SF spectra, statistical testing of link to measurement array size and TF) were carried out at the single participant level, so each participant comprises a test of the hypothesis under scrutiny. The sEEG contacts in the gray matter were re-referenced using the pooled average of the 50% lowest signal amplitude white matter contacts (*Mercier et al., 2022*). A typical participant's contacts are shown in *Figure 2A*. From the sEEG time series, complex-valued phase was analyzed at 34 frequencies, using short time series methods (two-cycle Morlet wavelets). Center frequencies ranged from 1.0 to 97.0 Hz, on a logarithmic scale. A segment of typical phase data with a center frequency of 9.2 Hz is shown in *Figure 2B*.

The highly irregular sampling grid provided by the sEEG limits the statements that can be made about the cortical phase dynamics. Due to the inherent noise of the spatial array, it was not possible to assess the smoothness of the cortical phase dynamics in the present research. We therefore consistently refer to the SF of the cortical phase dynamics and separately discuss implications for TWs. Assessment of the ubiquity of smooth gradients of phase, TWs, has been made elsewhere (*Alexander et al., 2016*; *Alexander et al., 2013*).

The present research used linear algebra techniques to empirically construct a basis with which to decompose the phase signal, following and extending previous techniques (*Alexander et al., 2006*; *Alexander et al., 2019*; *Alexander et al., 2016*). From this simpler set of empirical functions (*Zhang and Moore, 2015*), we are able to estimate the SF spectra of the cortical phase dynamics. The general strategy for empirical construction of a basis using SVD is illustrated in *Figure 2*. In the present research, the phase data at each TF was entered into an SVD. The left singular vectors (LSVs) from the SVD comprise maps of spatial covariances in phase. There are important mathematical relations between SVD and Fourier analysis, and for appropriate signals (see *Methods*), the SVD will empirically recover approximate sinusoidal basis functions.

In *Figure 3*, we present examples of the use of SVD to decompose signals into components approximating the Fourier basis. *Figure 3A* shows a one-dimensional random walk. *Figure 3B* shows how SVD decomposes this signal into pairs of sinusoids of increasing TF. These sinusoid pairs are quantitatively similar to the family of one-cycle Morlet wavelets (see *Shinn, 2023* for a mathematical proof). *Figure 3c* shows an image of waves of water. We encoded the image as 'time-series' of phase at the dominant vertical frequency, then analyzed the 'spatial' vectors of phase in the horizontal dimension using SVD. The LSVs are approximate sinusoids corresponding to the Fourier series in the spatial dimension. *Figure 4A* shows how the corresponding singular values of these LSVs (as amount of variance explained) can be read as a power spectrum, with each successively higher SF component explaining less of the variance in the phase encoded image. *Figure 3F* shows how these methods have been applied to MEG data (*Alexander et al., 2006*; *Alexander et al., 2016*; *Alexander et al., 2013*) to extract global TW components of the signal, and components at higher SFs. Here, successive groups of LSVs from the MEG phase data show increasing wavenumber (approximately k=1, k=2 …). *Figure 4B* shows the SF spectrum computed for each of these *wave maps* using the method presented in the current research. The sum of these separate spectra represents the SF spectrum for the phase at this TF in the MEG.

The irregular sampling of cortical spatial coordinates in the sEEG poses a difficult problem for standard Fourier analysis, even for specialized techniques targeting non-uniform arrays (*Margrave and Ferguson, 1997*; *Candes et al., 2008*; *Ying, 2009*; *Tong et al., 2003*). Here, we make use of SVD to empirically decompose the phase data, before using multi-scale differencing on each LSV to compute the SF spectrum. The primary reasons for using SVD were (1) to prioritize patterns of high spatial covariance and so exclude noisy phase estimates, and (2) an assumption that the LSV vectors represent Fourier-like components of the signal, being a combination of the smooth sinusoidal bases recoverable from regular measurement arrays (*Zhang and Moore, 2015*; and *Figure 3*) and low rank corrections to 'noisy' spatial sampling (*Ruiz-Antolín and Townsend, 2018*; see **Methods**). This assumption is supported via surrogate testing of our method using brain signals whose SFs are known (i.e. MEG).

The irregular sampling of cortical spatial coordinates inherent in sEEG was overcome by first decomposing the phase matrix (columns of contacts by rows of samples, at a given TF) into components that capture the largest portions of spatial covariance in the phase signal. The LSVs of the SVD (*Figure 2C*) were then further quantified using multi-scale finite difference methods to estimate the SF spectrum. The multi-scale difference methods consisted, first, of reorganizing the sEEG contacts into triplets corresponding to the vertices of approximately equilateral triangles within the cortical sheet. Between-contact distances were calculated as geodesic distances (*Margulies et al., 2016a*) in the cortical sheet to accurately reflect the underlying topology of the cortex. See *Figure 5*, *Figure 5— figure supplement 1* for examples of geodesic paths on the cortical surface. These triangular areas were used to calculate the rate of phase change across each region of the LSV of phase. These techniques enabled the SF spectra of the cortical phase dynamics to be quantified for cortical regions of varying area, and this was done at each TF.

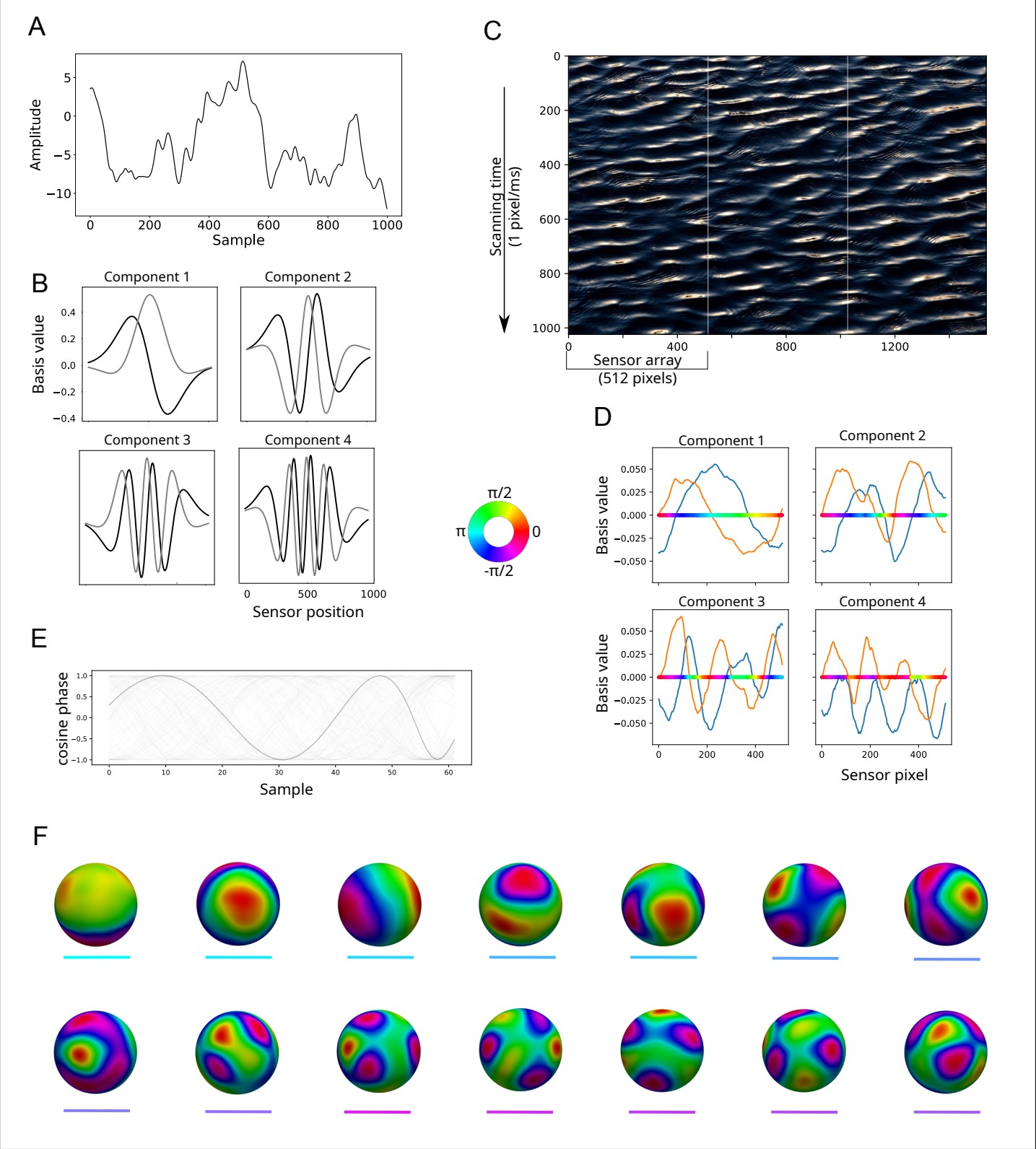

**Figure 3.** Relation between SVD and Fourier analysis. (**A**) A random walk of t time-steps is first filtered to remove very low frequency components, then successive segments (w=1024 samples) from the time-series are windowed with a Gaussian to produce a Hankel matrix $A_{wT}$, where T=t−w/2. Here, the one unwindowed sequence of the filtered random walk is shown. (**B**) The LSVs of the matrix $A_{wT}$ come in pairs, with each pair approximating the real and imaginary parts of a one-cycle Morlet wavelet. Here, the first four such pairs are shown, showing increasing frequency of the empirically derived

*Figure 3 continued on next page*

*Figure 3 continued*

Fourier components. The frequency of the first pair is closely tied to the size of the windowed sampling array. (**C**) Image of waves. The waves have a peak SF at 15 cycles per 1000 pixels vertically, and this dimension is treated as if it were time by scanning down the image (nominally) at 1ms per pixel. The waves in the image are aligned diagonally, moving towards the bottom right. The waves complete approximately one cycle every 512 pixels in the horizontal direction. This is the width chosen for the size of the measurement array, scanning the image vertically three times (white boundaries). The complex-valued phase at 15 Hz along the 'temporal' (vertical) axis is first calculated, and the phase dynamics along 'spatial' (horizontal) dimension are extracted using SVD. (**D**) First four components of the SVD of the vector of phases from image in (**C**). The real part of the component is shown in blue, imaginary part in orange. The phase angle corresponding to these real and imaginary values is shown by the colored horizontal line, with phase values shown on the color wheel. The first plot shows the dominant 'spatial' (horizontal) frequency, captured as a single cycle wave. Each successive component has an increment in the number of cycles represented, from the previous. (**E**) Time series of (the real part of) phase extracted from MEG data recorded with 151 magnetometers, here filtered at a center TF of 9.2 Hz. One sensor's data is highlighted in darker gray. (**F**) Wave maps (*Alexander et al., 2006*) extracted from the participant's phase data at 9.2 Hz. The MEG sensors are mapped to a sphere, and phase values are interpolated (values indicated by the color wheel). The head is viewed from the top, with the nose pointing left. The first three maps show global patterns of phase, with directions of flow characteristic of MEG data, namely left-right, inferior-superior, and anterior-posterior (*Alexander et al., 2006*; *Alexander et al., 2013*). These global patterns of phase are used to extract the dominant SF of TWs in the data (*Alexander et al., 2019*; *Alexander et al., 2016*). The components in this type of analysis are inherently spatio-temporal, since a spatial vector of smoothly changing phase implies a traveling wave over time. Successive groups of LSVs indicate components of the signal with increasing SF. (Colored lines below each map correspond to the spectra lines in *Figure 4B*).

The online version of this article includes the following figure supplement(s) for figure 3:

**Figure supplement 1.** Effects of bin size and calculation of spectral peak.

**Figure supplement 2.** Surrogate analysis of multi-scale contact arrays using random walk data.

## Single participant tests

We first quantified the estimated SF spectrum for all single participants, across all TFs. Two participants' results are shown in *Figure 5*. Lower TFs had more SF power than higher TFs (permutation test, p<0.05; *Panels A, D*). However, this is in large part to the better signal-to-noise ratio in the estimation of phase for the former (more time samples in the two-cycle Morlet wavelet) and similarly better signal-to-noise in signal measurement (higher signal amplitudes at low TFs). The effect is not necessarily a feature of higher TF signal per se. *Panels B and E* show the same spectra stacked over

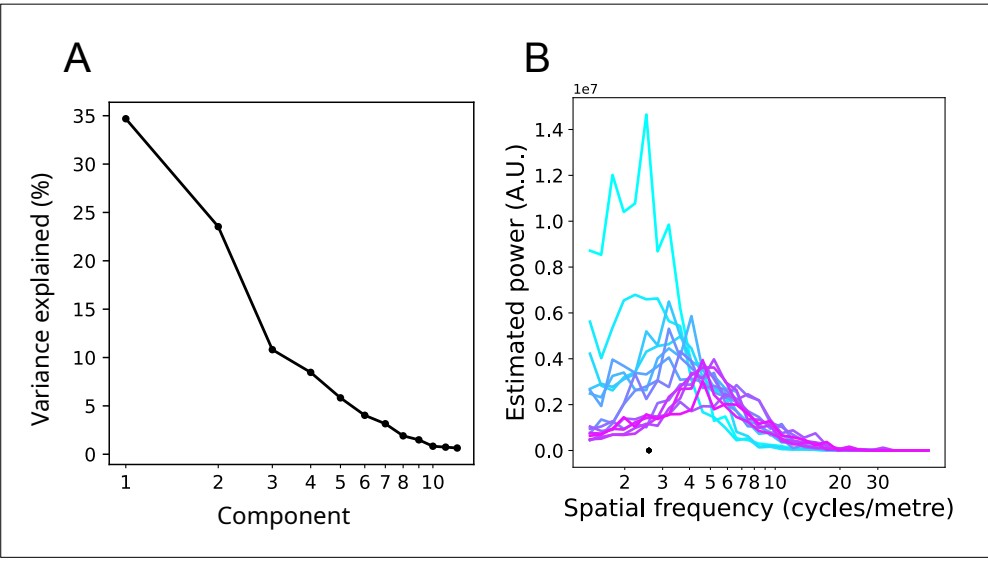

**Figure 4.** SVD and estimating spectral power. (**A**) Singular values obtained from SVD of the phase vectors from the image shown in *Figure 3C*. Because the first four components from the SVD are approximate sinusoids with discrete numbers of cycles, the Fourier spectrum can simply be estimated by reading off the singular values (shown here as % variance explained). The dominant SF is at wavenumber equals one cycle per 512 pixels (the size of the array), and the next highest SF components contribute successively less to the signal. (**B**) Estimated phase spectra for the wave maps of MEG data shown in *Figure 3F*, computed by the present method. The line colors correspond to the line labels in *Figure 3F*, and the black dot indicates the SF equivalent to the maximum distance between sensors. The spectra peak at a range of different SFs, with the global maps in *Figure 3F* showing the lowest SFs and highest power, and maps 6–14 showing the highest SF and least power.

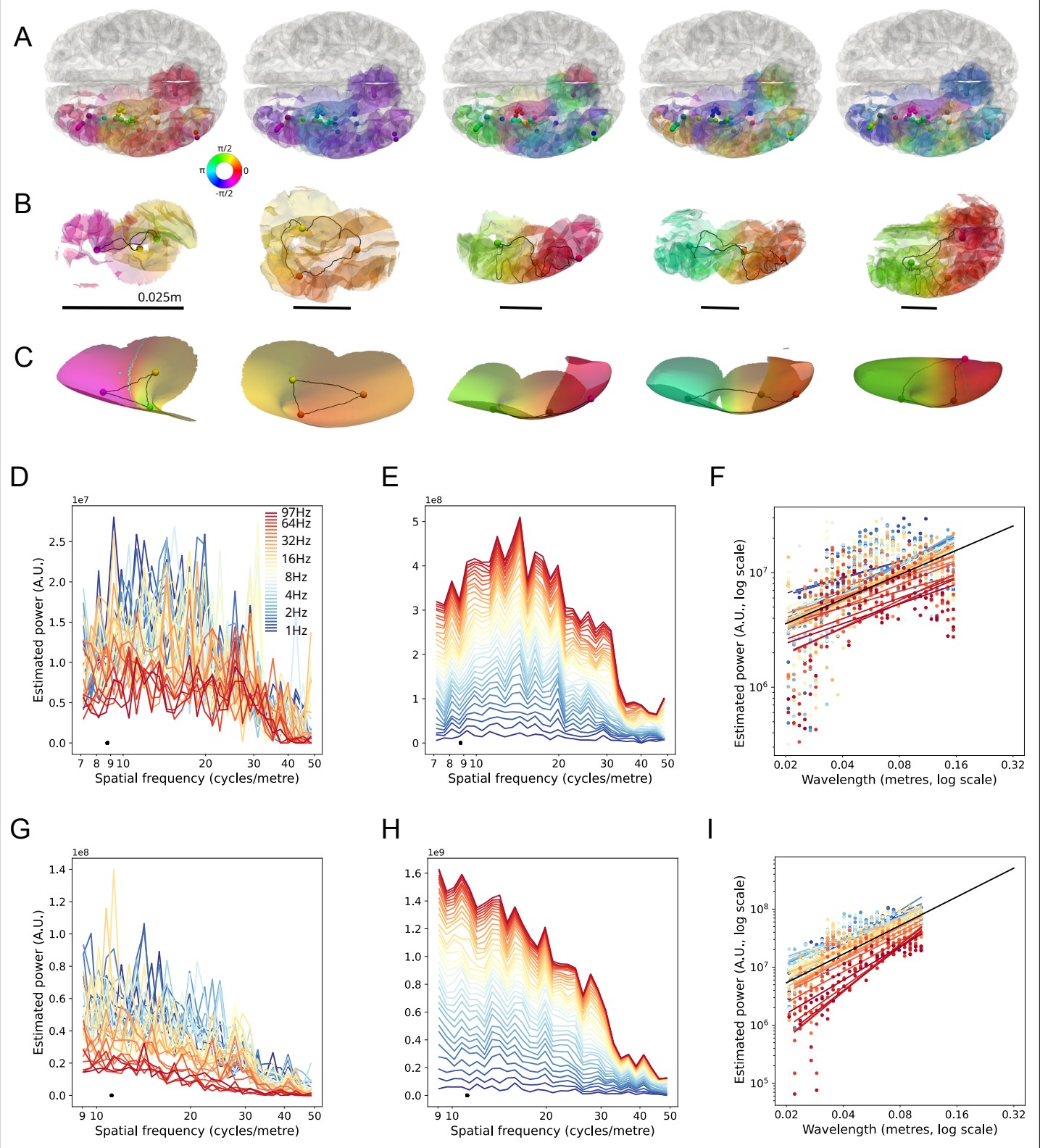

**Figure 5.** LSVs of sEEG participants, equilateral triangles of LSV phase and estimated SF spectra for two participants. (**A**). Cortical surface with phase values of the first five LSVs from a single participant at center TF 3.5 Hz. Gray matter contact locations are shown with spheres, and the phase value of the LSV at each site is shown on color scale. Phase values on the cortical surface are interpolated between contacts for visualization purposes. (**B**). Examples of approximately equilateral triangles of contacts, used to calculate the SF spectrum. The shortest geodesic distance between contacts is

*Figure 5 continued on next page*

*Figure 5 continued*

shown with the black line. Phase on the section of cortical surface is interpolated for visualization purposes. The scale of each equilateral triangle (square root of triangle area) is indicated with the black reference bar. (**C**). Same examples shown in (**B**), except projected onto the inflated cortex to show the unfolded triangle. (**D**) SF spectra for one participant, over TFs 1.0–97.0 Hz (color scale). The largest triangle size is indicated with the black dot. (**E**). Same as (**D**), except shown as a stacked spectrum. (**F**) Regression lines for (logarithm of) wavelength versus SF power at all TFs (colored lines), for the participant shown in (**D**) (only regression lines significant at p<0.05 shown). Spectrum values shown as dots, colours indicating TF. The mean regression line (mean of slopes and offsets of significant regressions) is shown with a black line and extended to 0.32 m wavelength for visualization purposes. (**G**) SF spectra for a second participant. (**H**) Same as (**G**), except shown as a stacked spectrum. (**I**) Regression lines for wavelength versus SF power (colored lines), for the participant shown in (**G**). Conventions the same as (**F**).

The online version of this article includes the following figure supplement(s) for figure 5:

**Figure supplement 1.** Examples of the equilateral triangles between contacts used in the multi-scale differencing.

**Figure supplement 2.** Stacked SF spectra, across TFs 1.0–97.0 Hz (color scale), for all participants.

**Figure supplement 3.** Regression lines for wavelength versus SF power at TFs 1.0–97.0 Hz (colored lines), for all participants.

TF and reveal the characteristic shape of the estimated SF spectrum. The spectra peak in power at a wavelength approximately equivalent to the size of the largest equilateral triangle defined on the measurement array, sometimes at a slightly lower SF. The spatial power of the phase dynamics decreased monotonically with increasing SF (permutation test over TFs, p<0.05). Within the range beginning at the maximum triangle size in the measurement array (equivalent to ~8.5 and 11 cycles/m in these cases), the spatial power drops steeply, before flattening at higher SFs.

This effect was statistically significant in 19 out of 23 participants (permutation test over TFs, p<0.05; see *Figure 5—figure supplements 2 and 3*). All participants' mean SF slopes trended in the same direction, with participants in the top 50%, by numbers of triangles, all being significant at p<0.001. This suggests a practical cut-off value for number of triangles in the present method of n=1500 for being able to construct sufficiently low noise spectra to indicate a significant trend of SF versus SF power. This equates to around 50 gray matter contacts. Only one of 782 distinct partici-pant × TF regressions was significant with a slope in the opposite direction. We therefore conclude that SF of phase had the highest power at the lowest SF. This main result is consistent with previous studies of SF of cortical phase dynamics: the spatial power decreases with increasing SF (*Barrie et al., 1996*; *Ramon and Holmes, 2015*; *Freeman et al., 2003*; *Freeman et al., 2000*). The present results extend the previously measured spectrum into scales equivalent to a large fraction of the cortical size.

Examples of the low SF phase dynamics are shown in *Videos 1–3*.

## Surrogate tests

We used surrogate methods to perturb the sEEG phase signals with artificial signals whose ground truth SF was known. This perturbation was done over a wide range of relative weightings between real and surrogate signals. If the method accurately measures SF, addition of the surrogate signals will induce a peak at the SF of the surrogate signal, increasing in relative amplitude as the surrogate signal amplitude is increased. Examples of the surrogate signal are shown in *Figure 6— figure supplement 1*. *Figure 6A* shows the results for one participant at 9.2 Hz, perturbed by a surrogate anterior-posterior gradient at three different SFs. The shift in the peak of the SF

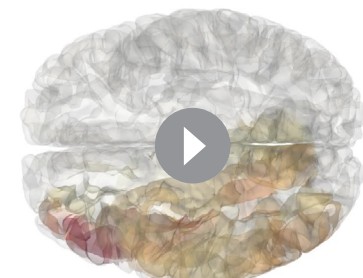

**Video 1.** Dorsal view of one participant's dominant spatial patterns of phase at 3.5 Hz. The video shows the time-series of a low-rank model of phase dynamics, constructed from the first three LSVs. Together, these LSVs account for 43.6% of the variance in phase. This low SF activity dominates the dynamics. The entire video comprises 6 s of cortical activity. Color values are cosine of the phase (yellow=-1, red = 1). Values between contacts are interpolated for visualization purposes. The anterior region of the cortex is to the left.

https://elifesciences.org/articles/100674/figures#video1

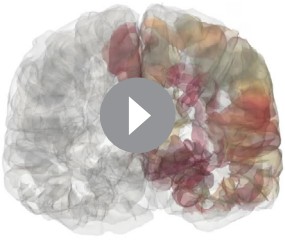

**Video 2.** Anterior view of one participant's dominant spatial patterns of phase at 8.0 Hz. The video shows the time series of a low-rank model of phase dynamics, constructed from the first four LSVs. Together, these LSVs account for 39.3% of the variance in phase. This low SF activity dominates the dynamics. The entire video comprises 6 s of cortical activity. Color values are cosine of the phase (yellow=-1, red = 1). Values between contacts are interpolated for visualization purposes. The left hemisphere of the cortex is to the left.

https://elifesciences.org/articles/100674/figures#video2

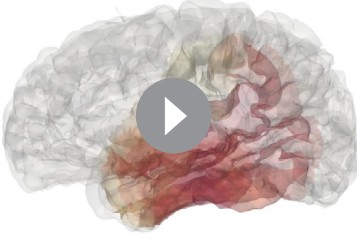

**Video 3.** Left view of one participant's dominant spatial patterns of phase at 8.0 Hz. The video shows the time series of a low-rank model of phase dynamics, constructed from the first three LSVs. Together, these LSVs account for 40.8% of the variance in phase. This low SF activity dominates the dynamics. The entire video comprises 6 s of cortical activity. Color values are cosine of the phase (yellow = -1, red = 1). Values between contacts are interpolated for visualization purposes. The anterior region of the cortex is to the left.

https://elifesciences.org/articles/100674/figures#video3

spectrum is apparent at low levels of injection (1/3 ratio) of the surrogate signal, consistently toward the SF peak of the surrogate. As the perturbation increases in weight, the surrogate contribution to the spectrum becomes clearer. The perturbation is approximately at the correct SF when twice the strength of the empirical signal (2/3 ratio) and remains so until ratio 7/8.

The surrogate SF signals were tested over 14 steps (at each TF) for a wide SF range (2 cycles/m to 16 cycles/m) and quantification showed the correct relation to the veridical SF. This was tested by regression of SF peak versus SF of surrogate (repeated for each surrogate direction and surrogate weighting) and was significant down to and including the 1/2 surrogate ratio (permutation tests over participants, p<0.001, ratio 1/3 not significant). Another example of surrogate tests on the same participant is shown in *Figure 6—figure supplement 1*, here with a left-right gradient imposed over the same same empirical phase signals. The Figure also shows cross-participant averages for these surrogate runs. Also shown are SF spectra with pure surrogate signal, giving a clear idea of the residual noise levels in the SF spectra estimation, since the only source of noise in this case is the irregularity of the sEEG array. Due to the highly irregular sampling of the cortex, the exact locations of the peaks for the veridical SF surrogate varied slightly from participant to participant for the same surrogate SF, but always trending within each participant in the correct ordering of SF peaks (peak SF vs. surrogate SF, permutation test over participants, p<0.001).

We also checked the accuracy of SF estimation by using the technique on neurophysiological data with known SF characteristics, here MEG. The method successfully recovered the distribution of SF power previously reported for phase signal (*Figure 6B*; compare Figure 2 in *Alexander et al., 2016*, Figure 2 in *Alexander et al., 2019*). Additional checks were made on the method using MEG data. *Figure 4* shows that each of the individual LSVs from SVD on MEG phase had different SF peaks using the method when computed separately, as expected from visual inspection of the LSVs (*Figure 3F*). This confirms that the peak SFs were being detected accurately, as the LSVs changed in SF characteristics. *Figure 6B* indicates how these separated curves sum to produce the completed SF spectrum in MEG.

However, we wanted to definitively rule out that the method was just constructing distributions with left-pointing tails, and that we were quantifying the left-pointing tails as ~1 /f spectra.

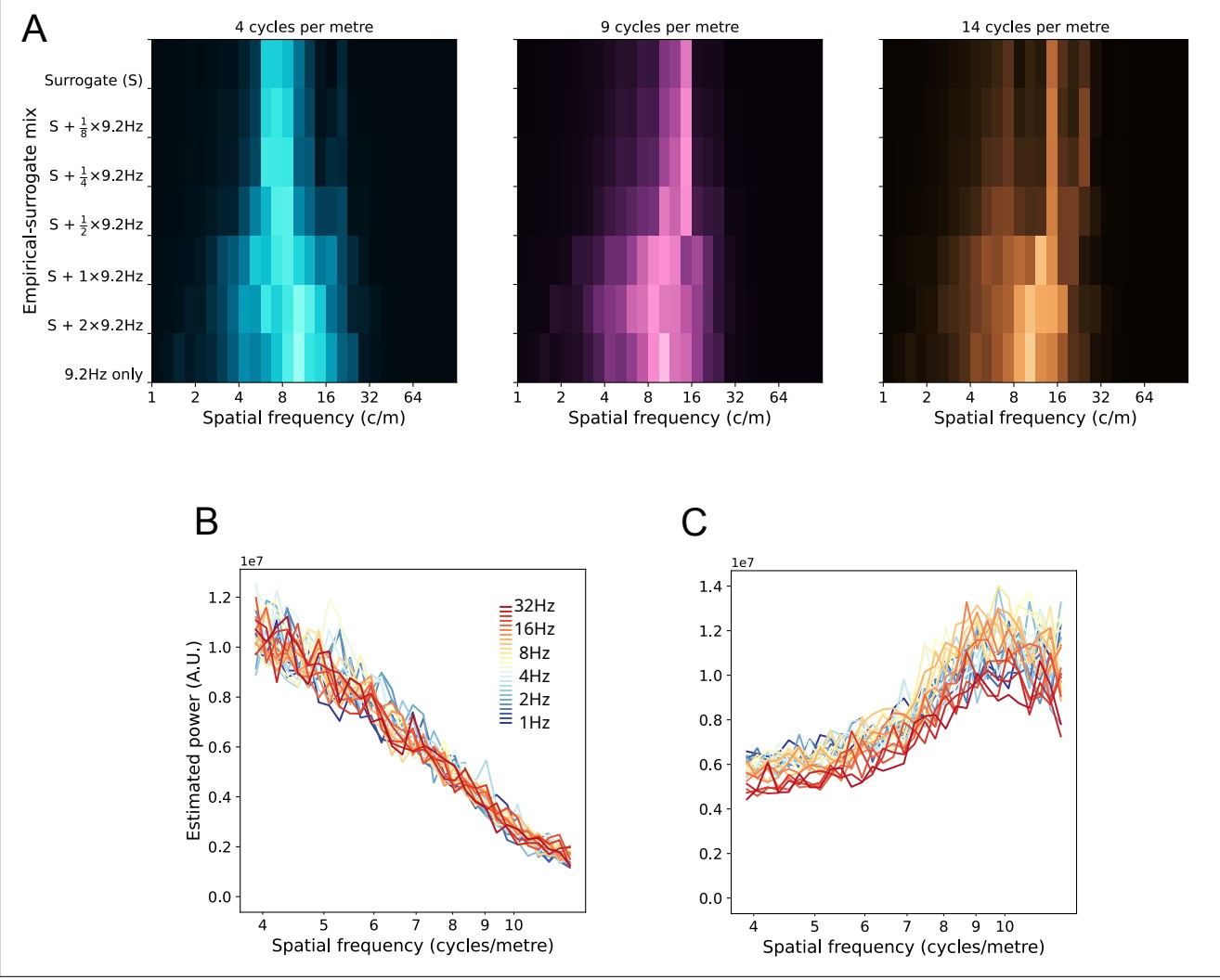

**Figure 6.** Surrogate tests on individual participants. (**A**) Effects of injection of veridical SF surrogate signal on the empirically derived spectrum. The left panel shows the spectrum of a pure surrogate signal (**S**) with an anterior-posterior gradient and a SF at 4 cycles/m (top row within plot). The empirical spectrum for the phase of this participant at a center frequency of 9.2 Hz is also shown (bottom row within plot). The effects of different weightings of empirical and surrogate signals vary vertically over the plot. The middle panel follows the same conventions, except a surrogate signal at 9 cycles/m is shown. The right panel shows the effect of a surrogate signal at 14 cycles/m. (**B**) SF spectra of phase for a participant with MEG measurements. Each spectrum shows the characteristic monotonic decrease in power with increasing SF. This data is from the same participant as *Figure 4B*, where the unsummed spectra for each LSV at 9.2 Hz are shown. (**C**) SF spectra for surrogate MEG measurements. Here, the singular values of the first 28 components are reversed in order before reconstructing the phase time series, creating phase signal in which high SFs dominate. The plot shows that the present methods successfully characterize the surrogate signal as SF spectra with slope reversed from the empirical signal.

The online version of this article includes the following figure supplement(s) for figure 6:

**Figure supplement 1.** More surrogate testing examples.

**Figure supplement 2.** Effects of randomly removing sensors from a semi-regular array.

**Figure supplement 3.** Surrogate testing for the effect of average reference and removal of DC component on the phase spatial spectrum.

**Figure supplement 4.** Illustration of the removal of negative SF components.

To this end, we constructed an MEG time series of phase in which the high SFs dominated. This was achieved by reversing the order, and so the weighting, of the first n=*28* singular values and reconstituting the phase matrix according to *Equation 2* (*Methods*) and as described in *Figure 2C*. *Figure 6C* shows that this surrogate data had a SF spectrum that increased monotonically with SF over the relevant range, that is the method accurately characterized the new SF spectrum.

To additionally rule out whether the produced SF spectra were an artifact of the sparsity and irregularity of the sEEG array, we produced MEG time series of phase in which the sensors were randomly removed until the phase vectors were only one-third their normal length, that is 51 sensors from an original 151. This procedure was repeated for several random removal sequences. The results are shown in *Figure 6—figure supplement 2* and demonstrate that the effect of increasing measurement array sparsity and irregularity was to increase the noise of the SF estimate, as expected, but did not alter the overall shape of the spectra (wavelength vs. power, permutation test over runs, p<0.001). This procedure was repeated for the MEG surrogate in which high SFs dominated (i.e. SF power increasing with SF), with the same conclusions (wavelength vs. power, permutation test over runs, p<0.001).

We also tested for the effects of the average referencing used on the sEEG data (see *Methods*). Global synchrony appears as the DC component (SF = 0) in spectra of phase. Computing an average reference increases the global synchrony of phase in an artefactual fashion. While this component is well separated from the minimum of the SF range of interest here, we needed to rule out the influence of the referencing method on our estimated spectra. To this end, we produced spectra for MEG data, for which the case of a zero weight 'average reference' is also available (i.e. no reference). We compared spectra with an average reference added at increasing levels of influence. Increasing the weight of average reference up to a factor of eight times the usual average reference did not disturb the conclusions of this report (see *Figure 6—figure supplement 3*). This procedure was tested with the methodological step that removed negative SF components (see *Methods*) and without. The results showed that this step both lowers the DC component of the SF phase spectrum—and thereby ameliorates the effects of any artefactual global synchronous component—but also does not introduce a false maximum at the relevant low SFs due to lowering of the DC component.

## Cross-participant analysis

We analyzed the relation between wavelength and estimated power for each participant using linear regression. The regression lines for the logarithm of wavelength versus estimated power are shown in *Figure 7*, for all individual participants at 9.2 Hz (permutation test over participants, p<0.001). The regression lines for all TFs are shown in *Figure 7—figure supplement 1* and all TFs showed a significant decrease in SF power with SF (permutation test over participants, p<0.001). The participants with the fewest triangles did not always show significant individual wavelength by SF power regressions at a given TF, but the regression lines almost universally trended (except one case) in the predicted direction. The present results extend the previous finding of ~1 /f spectra in cortex (*Freeman et al., 2000*; *Llinás and Ribary, 1993*) to TFs up to 100 Hz.

The cross-participant analyses also confirmed that peak SF power declined linearly with increasing TF (logarithmic scale). This is shown in *Figure 7D* for the participant-aggregate spectra (permutation test, p<0.001) and for the individual participant's spectra in *Figure 7E* (permutation test over participants, p<0.001). This effect may be wholly or in part due to measurement and quantification noise at higher TFs, as previously mentioned. Importantly, the cross-participant analyses revealed no effect for TF on SF of peak SF power. This lack of effect obtained for the participant aggregate spectra (*Figure 7D*) and for the permutation test over all individual participants' regressions of (logarithm of) wavelength at peak SF power (*Figure 7F*). This means that the relative shapes of SF spectra were the same, regardless of TF. That is, the SF spectral power decreased monotonically over the relevant range from 8 to 12 to 50 /cm, for TFs 1.0–100 Hz.

The linear regressions we performed excluded SF estimates lower than the reciprocal of the maximum triangle size. This decision arises from considerations of Fourier analysis, in which the maximum wavelength detectable in broadband signal is a strict function of the maximum range of the (regular) grid size. In our results, the apparent SF power drops off for SFs lower than this, but this part of the spectrum is not meaningful. With the present method, maximum array size was weakly correlated with the SF at peak power for each participant (peak SF in TF aggregate spectrum vs. maximum inter-contact distance, permutation test, p<0.05, *r*=0.31). Maximum array size is defined by the two most (geodesically) distant contacts. It does not, therefore, always reflect the typical largest distances for the array. We describe in the *Methods* how peak SF is a function of the distribution of triangle sizes in irregular arrays while using the present SVD methods. For this reason, we use maximum triangle size as a bound for SF power regressions.

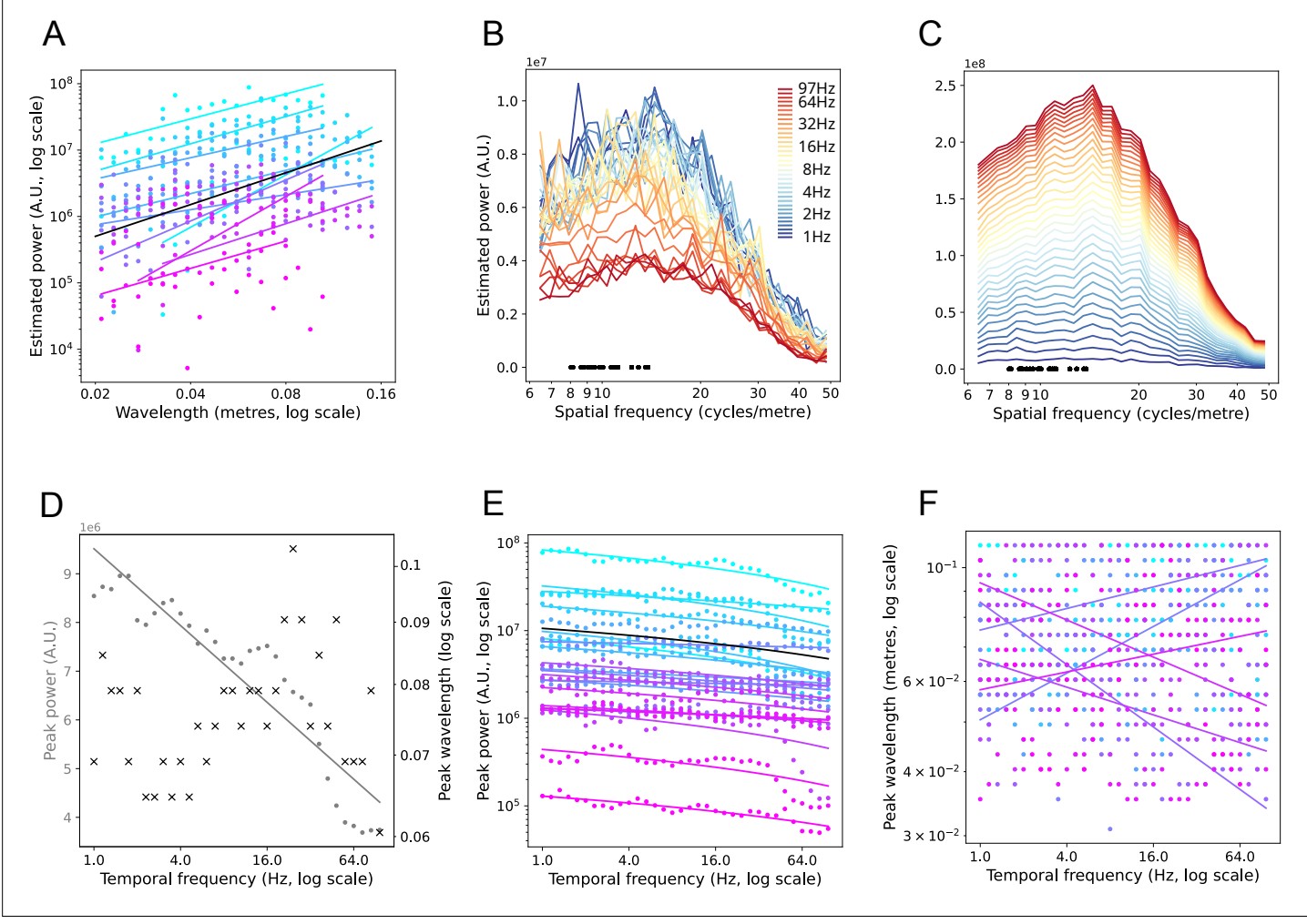

**Figure 7.** Cross-participant analyses. (**A**) Regression lines, for all participants, for wavelength versus SF power at center TF 9.2 Hz. Participants are ordered from most number of triangles (n=11051; light blue) to least number of triangles (n=56; lilac). Spectra values are shown as colored dots (data from *Figure 5—figure supplement 2*). Only individual regressions at 9.2 Hz that were significant at p<0.05 are shown as regression lines. The mean slope and offset (of only significant regressions) is shown as a black line. (**B**) Participant aggregate spectra for all TFs. TF is shown on the colored scale. Individual lines are the mean spectra over 23 participants at that TF. The maximum triangle size (root area, expressed as equivalent SF, i.e. 1 /m) is shown for all 23 participants, as black dots. (**C**) The same spectra as (**B**), except shown as a stacked histogram, giving the overall shape of the spectra. (**D**) Regression of peak power versus TF (gray dots) and SF at peak power versus TF (black crosses; data from B). Significant regression line for peak power is shown in gray. (**E**) Single participant regression lines for peak power versus TF. Color conventions the same as (**A**), and the mean slope and offset of the significant regressions are shown as a black line. Peak power is shown on a logarithmic scale for visualization purposes only. (**F**) Single participant regression lines for SF at peak power versus TF. Color conventions the same as (**A**).

The online version of this article includes the following figure supplement(s) for figure 7:

**Figure supplement 1.** Cross-participant analyses by TF.

The array sizes used here did not reach the full extent of the cortex (~45 cm); however, we were at the outset interested in the peak SF for the case where the array size extended to the entire cortex. For visual convenience, the regression lines in *Figure 5*, *Figure 5—figure supplement 3* are extended to larger cortical distances. The linear relationship suggests that triangle sizes extending to the full range of the cortical sheet will result in peak SF estimates larger than the maximum reported for the present participants. This prediction is supported by previous analyses using EEG and MEG, showing that the peak SF coincides with the size of the whole head array (*Alexander et al., 2016*; *Freeman et al., 2003*), which in turn coincides with the entire length of the cortex accessible over the upper half of the skull. It would be informative to repeat the analyses presented here after careful selection of sEEG participants from all available databases for maximum cortical sampling extent, or as technical

advances in cortical sampling further improve. We expect that participants with greater than 30 cm coverage will extend the linear trend reported here.

## Effects of TF power

The main analyses all used phase as the input variable, reflecting our interest in cortical TWs. However, the main conclusions are not disrupted when TF power is included in the calculations. *Figure 8* shows the results when TF power is normalized on a contact-by-contact basis (see *Methods*). Individual participant spectra are largely unchanged, with a suggestion of a weak increase in power at higher SFs (>30 cycles/m) compared to lower SFs (<10 cycles/m). *Figure 8—figure supplement 1* shows the results of including (unnormalized) TF power. The clearest effect is the influence of decreasing TF power on SF power as TF increases (panels *A* and *B*) – TF power is small at higher TFs. We visually correct for this effect by plotting the relative values within the SF spectrum at each TF. Panels *C* and *D* show there is an apparent increase in SF power between 12 and 20 cycles/m compared to the phase-only analyses, but not sufficient to overshadow the SF components 8–12 cycles/m. It is likely that much of this effect is due to noise in the measurement of signal amplitudes from contact to contact, since it largely disappears when power is normalized on a contact-by-contact basis. If a sensor array (e.g. a camera pixel array) is more sensitive at some sites than others, this will appear as an artefactual increase in higher SF components of the image.

## Discussion

In this research, we aimed to quantify the low SF part of cortical phase dynamics spectrum, up to the range of large-scale TWs (wavelength >8 cm), using cortical depth electrodes. We developed a novel method to fill this important gap in our understanding of macroscopic phase dynamics, addressing a current limitation in the literature. Previous estimates have either relied on extracranial measurements (*Alamia et al., 2023*; *Alexander et al., 2006*; *Alexander et al., 2016*; *Ito et al., 2005*; *Klimesch et al., 2007*; *Massimini et al., 2004*) or used smaller measurement arrays (≤8 cm; *Zhang et al., 2018*; *Barrie et al., 1996*; *Woolnough et al., 2022*; *Eckhorn et al., 2004*). This means either the results were biased by a low pass filter (due to volume conduction in EEG or distance to measurement array in MEG), or had a maximum measurable SF below the large-scale range, respectively. We show that the spectral power increases with wavelength, up to the limit imposed by the maximum distances in the recording array. The present results are therefore consistent with the existence of large-scale TWs in the cortex (*Figure 1D*) and are inconsistent with the alternative hypotheses for the measurement of these waves extracranially (*Figure 1E & F*). Low SF activity dominates the cortical phase dynamics; most of the spatial power is at the longest wavelengths. In other words, most of the spatial power measured extracranially is therefore produced by long wavelength activity in the cortex.

Considered from the perspective of locally measured phase—that is local to a single gray matter contact—the measured SF spectra imply that most of the variance in phase is a function of low SF phase dynamics. Note that this statement does not include cross-frequency effects, outside the (relatively broad) range of the frequency bandwidth of the phase estimator used in this study. The present result partly answers the longstanding question of why there is so much variability in locally measured cortical responses, even to the same stimulus (*Zhang and Moore, 2015*; *Arieli et al., 1996*, cf. *London et al., 2010*). We suggest that low SF phase dynamics index a distinct scale of cortical activity and, judged from their dominance of that activity, comprise an important scale for understanding cortical function (*Freeman, 2015*). The variability is a mystery if the response in a functionally relevant area is considered in the absence of the global phase dynamic context. Our results suggest the variability can be controlled for if the global context is also measured.

The shape of the SF spectrum in the range ~8–50 cycles/m is therefore consistent with reports of the spectrum at smaller cortical scales (*Barrie et al., 1996*; *Ramon and Holmes, 2015*; *Freeman et al., 2003*). The cortical SF spectrum is generally described as $1/f^{\alpha}$, where the exponent $\alpha$ is a function of the type of measurement (e.g. gray matter, cortical surface, or extracranial; *Barrie et al., 1996*; *Freeman et al., 2003*; *Freeman et al., 2000*). Since peak SF also increases with the total size of a participants' sEEG array (maximum inter-contact range), we assume that the $1/f^{\alpha}$ distribution of SF extends to the maximum size of the human cortex; this is approximately 45 cm linear distance within the unfolded gray matter. This would be consistent with spectral peaks at lowest SFs measured using

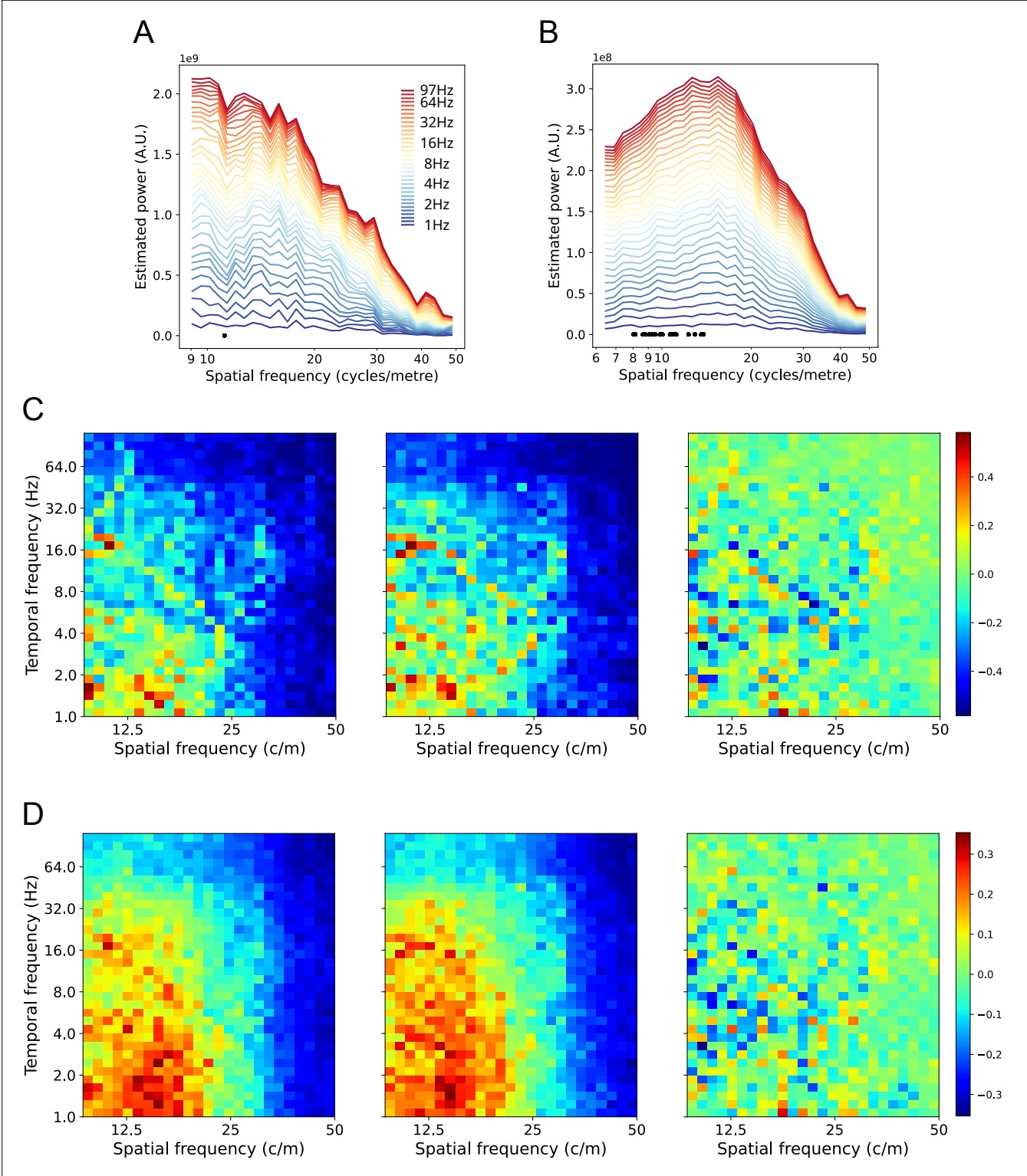

**Figure 8.** Effects of including normalized TF power on SF spectra. (**A**). Participant's stacked SF spectra with normalized TF power included in the estimate. TFs are shown on the color scale. Compare with the same participant, phase-only spectra, *Figure 5H*. (**B**). Aggregate stacked SF spectra with normalized TF power included, over 23 participants. Compare with the aggregate stacked spectra shown in *Figure 7C*. (**C**). Participant's SF spectra as SF by TF plot. The left plot shows the SF spectra with normalized TF power included (data taken from A). Power values are shown as colors, normalized between 0 (cold) and 1 (hot). The middle plot shows the phase only SF spectra (data taken from *Figure 5H*), normalized between 0 (cold) and 1 (hot).

*Figure 8 continued on next page*

*Figure 8 continued*

The right plot shows the difference between the two SF spectra, with differences given on the color scale. (**D**). Aggregate spectra for 23 participants, conventions the same as C. Data are taken from (**B**) and *Figure 7C* for the left and middle plots.

The online version of this article includes the following figure supplement(s) for figure 8:

**Figure supplement 1.** Effects of including TF power on SF spectra.

EEG (*Ramon and Holmes, 2015*; *Freeman et al., 2003*) and MEG (*Alexander et al., 2016*), since this is what is expected when a low pass filter is applied to a signal dominated by low SFs.

Volume conduction in the cortex acts as a band-pass filter, attenuating signals at higher SFs. Volume conduction effects have been quantified in the gray matter, for local field potentials, electrocorticogram, and in vitro (*Kajikawa and Schroeder, 2011*; *Nelson et al., 2013*; *Dubey and Ray, 2019*; *Orczyk et al., 2021*). These effects are less than 1 cm in range and so smaller than the minimum 2 cm range assessed here. These effects cannot explain the distribution of SFs between ~8 and 50 cycles/m, since they can only attenuate SFs >100 cycles/m.

A surprising finding was that the shape of the SF spectrum did not vary much with TF. This finding implies that low SF cortical phase dynamics arise at TFs up to 100 Hz, the maximum TF tested here. Some care should be taken in interpreting the SF spectra at TFs above 60 Hz, since band-pass filtering alone is not sufficient to remove the low TF components of spiking activity (*Zanos et al., 2011*).

Gamma (30–60 Hz) and high gamma (>60 Hz) synchrony have been functionally implicated across a range of scales in the cortex. Several studies have quantified the functional role of gamma band oscillations within cortical areas, notably in the visual and motor areas. Spike-triggered analysis of local field potentials in V1 has shown that TWs have a greater range of propagation under low-contrast visual stimulation, compared to high contrast (*Nauhaus et al., 2009*). In multiple visual areas, *Womelsdorf et al., 2007* demonstrated that gamma phase synchrony between recording sites precedes high amplitude correlations between those same sites. V1 oscillations were analyzed in the range 50–80 Hz using the transfer entropy of the phase and showed the activity to be compatible with directional wave propagation (*Besserve et al., 2015*). *Blakely et al., 2014* analyzed temporal covariance of gamma band activity recorded in ECoG arrays to map finger flexions in the motor cortex.

At a larger spatial scale, the (putative) causal influence of frontal regions on more posterior zones has been demonstrated, using the phase slope index, across a wide range of TFs, including gamma (*Casimo et al., 2016*). sEEG recording has been used to analyze long-range high-frequency phase synchronization (*Arnulfo et al., 2020*). Using zero- and nonzero-lag phase measures, these authors demonstrate that phase locking occurs in the high TF ranges 100–450 Hz, across globally ranged cortical regions. None of these studies, however, specifically assessed the SF spectrum of the phase signals.

The existing literature on gamma-band TWs is patchy but consistent with our findings here. Macroscopic TWs are commonly reported in the delta to beta ranges (*Alamia et al., 2023*; *Alexander et al., 2006*; *Ito et al., 2005*; *Alexander et al., 2013*; *Sauseng et al., 2002*). However, TWs have also been reported for low gamma frequency (40Hz) using MEG, in an event-locked paradigm (*Llinás and Ribary, 1993*). Similarly, EEG-based estimation of SF has been reported for TFs from 7 to 63 Hz, with similar spectra ($1/f^\alpha$) for SFs over this entire range (*Freeman et al., 2000*).

We outlined in the introduction the theory of cortical TWs as group waves (*Nunez, 1974*; *Wright et al., 2001*; *Nunez and Srinivasan, 2006*), which means the group wave velocity can exceed the speed of information propagation. Specifically, the rise and fall of coordinated cortical activity can outpace axonal conduction times, since it depends on patterns of spikes that have previously arrived (*Skarda and Freeman, 1987*). If TWs are group waves and do not directly reflect the speed of information propagation, this may appear to support the argument that the cortex's electric fields are epiphenomenal, and that spiking activity of individual neurons is the ground truth. However, the literature on TWs shows that they can influence spiking patterns of individual neurons (*Woolnough et al., 2022*; *Zanos et al., 2011*; *Takahashi et al., 2015* cf. *London et al., 2010*). To this, we may add reports of the endogenous electro-magnetic field altering firing patterns of neurons (*Fröhlich and McCormick, 2010*; *Anastassiou et al., 2011*). It seems likely that a major driver of cortical plasticity is precisely the dominant SFs found in the cortex, since post-synaptic membrane depolarization is the second major ingredient of long-term potentiation (*Nicoll, 2017*). An evolutionary argument can also be applied, since dominant low SF phase dynamics will be more metabolically expensive than otherwise

quiescent, mostly locally activated cortex. This implies critical function rather than epiphenomenon. Taken together, these arguments suggest that the pattern of phase organization in the cortex has a causal role and is not the epistemological equivalent of heat generated by a CPU.

The present results have important methodological implications. Considerations from Fourier analysis raised in the introduction indicate that the maximum spatial sampling range is critical to setting the lower bound of detectable SFs. This would not be the case if the SFs were pure and isolated, but for broadband, highly mixed signals, it is. These considerations are supported empirically by the present results. The peak SF increases in wavelength with maximum contact distance for the participants (here, 0.12–0.24 m), which in practice is related to the number of contacts and maximum triangle size. In addition, we showed that there was a linear relation between peak SF and the triangle size used to estimate SF. The present findings also explain why detected TWs, regardless of cortical scale, generally have a wavelength ~2π (*Alexander et al., 2013*; *Ermentrout and Kleinfeld, 2001*). If the spatial spectrum is monotonically decreasing in power over the entire large-scale to microscopic range, then the highest power waves detectable on the array will be those with a wavelength approximately the size of the array.

We conclude that reports of TWs should include estimates of SF to allow or disallow comparisons between studies with different extents of cortical coverage, since they may be accessing different SF ranges and therefore different functional connectivity ranges (c.f. *Woolnough et al., 2022*; *Rubino et al., 2006*). The present results, along with considerations from Fourier analysis, indicate that only methods that have large-scale coverage of cortical activity can assess the dominant scale of cortical phase dynamics. Unfortunately, source localization techniques, designed to improve upon sensor-level measurements, remove long-range spatial correlations in the signal; both artefactual and real (*Srinivasan et al., 1998*; *Hindriks, 2020*). This excludes these techniques from accurately estimating the SF spectra of phase dynamics since they filter out precisely the most dominant components, according to the present results. Similar considerations apply to methods to remove volume conduction effects from local field potential recording, such as taking the first- or second-order spatial derivatives.

For this reason, we argue, forward models of cortical activity are better suited to discovery of cortical dynamics from extracranial measurements (*Grabot et al., 2025*; *Kupers et al., 2021b*; *Kupers et al., 2021a*). The present results suggest that the key methodological assumption of source localization techniques, namely that localized peaks in activity can be quantified to produce a mapping from extracranial patterns of activation to cortical locations, may not be sustainable. The contrary case, that high SF peaks exist in the cortical spectrum in real time (or that the low SF dominance is functionally irrelevant) should be demonstrated rather than implicitly assumed. Worse, confirmation of our present results would more broadly suggest that the endeavor itself is less meaningful than previously appreciated. If high SF activity forms only a small part of the total SF spectrum, then source localization is not just an ill-conditioned problem, but perhaps not the problem. At the least, estimates of relative amounts of cortical activity explained by sources thus localized should be attempted (see *Piastra et al., 2024*; *Chaumon et al., 2021* for some discussion of these issues).

TWs have been shown to allow prediction of future phase at single recording sites two wave cycles into the future. This effect was shown for both macroscopic MEG and ECoG (~8 cm range) measurements (*Alexander et al., 2019*). The ability of TWs to predict future local activity shows that the long-range spatio-temporal correlations inherent in the structure of low SF activity are functionally significant. Future research could extend this approach using high temporal resolution sEEG to detect individual spikes and to characterize large-scale phase dynamics and to predict the former from the latter. More generally, most functionally related results concerning large-scale TWs have been obtained using extracranial methods (*Ito et al., 2005*; *Klimesch et al., 2007*; *Massimini et al., 2004*; *Sauseng et al., 2002*; *King and Wyart, 2021*). The reported linkage between late event-related potentials and TWs (*Alexander et al., 2006*; *Alexander et al., 2013*; *Alexander et al., 2008*; *Alexander et al., 2009*) means the body of research into late event-related potentials can guide experimental verification of the functional significance of macroscopic TWs at the single trial level.

Our conclusions regarding the dominance of large-scale cortical activity are consistent with other recent findings in neuroscience. There has been a resurgence of interest in the global structure of the cortical functional map, driven by developments in fMRI, large-scale open-access databases, and numerical analysis techniques (*Pang et al., 2023*; *Huntenburg et al., 2018*; *Margulies et al., 2016b*). An intrinsic functional coordinate system has been proposed: along with hemispheric lateralization,

a global coordinate axis has been described in terms of a visual to somatosensory/motor dimension, beginning at the occipital pole and calcarine sulcus, extending around the cortical hemispheres through the temporal lobe, frontal areas, and to the central sulcus. Another dimension, at approximately twice the SF, is a function of the cortical distance between the primary areas and transmodal areas (*Huntenburg et al., 2018*). Eigenmode analysis of fMRI data shows that macroscopic wavelengths dominate the spatio-temporal activity at time scales characteristic of fMRI (*Pang et al., 2023*). This is consistent with previous findings in the EEG and MEG showing that long wavelengths are ubiquitous features of the spatio-temporal activity in the ranges delta to beta (*Ito et al., 2005*; *Alexander et al., 2013*; *Woolnough et al., 2022*; *Mohan et al., 2024*; *Alexander et al., 2015*). Here, we have shown that low SF phase dynamics dominate the spatio-temporal organization of cortical activity measurable in the gray matter, and that these global dynamics extend to fast cortical time scales into the high gamma range, at 10's of milliseconds.

## Methods

### Data acquisition

We summarize the relevant details of the RAM data set here (https://memory.psych.upenn.edu/RAM_Public_Data). Full details of data acquisition are reported elsewhere (*Ezzyat et al., 2017*). 35 human participants were part of a multi-center sEEG monitoring study. The participants were drawn from experiments 'FR5' and 'catFR5', and we included those that had complete data sets for present purposes (time-series, contact locations, cortical surface meshes). From the initial pool of participants, 23 had sufficient numbers of gray matter contacts to enable further analysis (see Numerical methods for criterion). Mean age was 34 years, nine female, four left-handed.

The participants' task was designed to assess the effects of electrical stimulation on memory-related brain function. Each participant engaged in a delayed free recall task in which they studied lists of words for a later memory test. Trials with stimulation were not used in the present study, and our interest was in the general characteristics of cortical activity, not especially dependent upon this task. The research protocol was approved by the institutional review board at each hospital, and informed consent was obtained from each participant. Electrophysiological data were collected from electrodes implanted within the brain parenchyma. Participants had temporal lobe depth electrodes, with frontal and parietal regions targeted as needed to best localize epileptogenic regions. Contacts on each electrode had a 5 mm inter-contact spacing. The sampling frequency was 1000 Hz, except for eight participants, for which it was 500 Hz. The number of gray matter contacts ranged from 31 to 108 with a mean of 58.

The data for each participant was denoised by first calculating the standard deviation of signal over samples, for each trial and sensor combination (*Mercier et al., 2022*). If the mean standard deviation for a trial was greater than ±3 the mean standard deviations from all trials, then that trial was removed from further analysis. Then, if the mean standard deviation (after recalculation) for a sensor was greater than ±1.5 mean standard deviations from all sensors, it was removed from further analysis. Finally, if the standard deviation (after recalculation) for a trial by sensor combination was greater than ±3 standard deviations from all trial by sensor combinations, it was replaced with the average signal (average calculated after excluding all over-threshold trial by sensor combinations).

Using the automated contact labeling supplied within the dataset (checked and corrected manually by clinicians), contacts were partitioned into three groups: gray matter, white-matter low-amplitude, and other white matter. We defined *white-matter low-amplitude contacts* as those whose standard deviation of signal amplitudes was in the bottom 50% of white matter contacts. The white-matter low-amplitude contacts were pooled to create an average reference for the gray matter contacts (*Mercier et al., 2022*). All numerical procedures were trialed using notch filters to remove the 60 Hz line signal and its harmonics and repeated without notch filters simply by avoiding these bands. No numerical differences were obtained for the common frequencies in the two versions of the data analyzed, so the former analyses are reported here.

### Numerical methods

#### Phase estimation

Phase is represented as a unit-length complex number, $\phi$.

$$\phi = e^{-i\theta}; \left\{ \theta \in \mathbb{R} | \text{-}\pi \leq \theta < \pi \right\} \tag{1}$$

We used complex-valued phase (*Alexander et al., 2019*; *Muller et al., 2021*) in conjunction with the complex SVD to avoid the (error) noise introduced by spatial unwrapping phase when using real-valued techniques (c.f. *Zhang et al., 2018*; *Alexander et al., 2006*; *Burkitt et al., 2000*; *Spagnolini, 1995*; *Margrave and Ferguson, 1997*; *Fisher, 1993*). Phase is calculated from the sEEG time-series, for a particular TF, using short-time-window Morlet wavelets (two cycles, *Herrmann et al., 2005*). Short time series methods were used because previous research has shown TWs to often last only one or two cycles, and windows of five or seven cycles can wash out the detail of fast temporal changes (*Alexander et al., 2006*). Use of short-time-window Morlet wavelets increases the temporal resolution of the phase estimate, at the cost of a decrease in TF resolution. For example, the wavelet with center TF at 4.5 Hz has a half-power bandwidth spanning 2.5–6.8 Hz. This lower TF resolution has positive consequences, since it makes the analyses herein more robust to small variations in TF across areas and regions.

We estimated phase for 34 center-TFs, logarithmically spaced from 1.0 to 97.0 Hz. Due to the low TF resolution of the two-cycle Morlet wavelets and number of frequencies extracted, this amounted to an oversampling of the frequency range. We use indices of $s$ and $t$ to refer to measurement sites and sample times, respectively, giving a matrix of complex-valued phases for each TF, $\Phi_{st}$.

Because Morlet wavelets have poor frequency resolution, the very short time-window Morlet basis is only approximately sinusoid and robust to relatively wide variations in TF (*Huang et al., 1998*) and violations of sinusoidal shape. In practice, any surge of activity up and down with a time course near the center TF will have a high correlation with the applied Morlet basis. We are therefore confident that the present results are not confabulated by 'ringing' effects (*Cohen, 2014*).

A potential criticism of the approach used here is that we estimate SF spectra for phase, while discarding amplitude information. The primary reason is our interest in TWs, which are generally defined in terms of phase. It is worth noting that single trial phase is a stronger determinant of mean signal amplitude than is single trial amplitude (*Alexander et al., 2013*). This has been shown for EEG, MEG, and ECoG. In other words, aggregate measures such as event-related potentials, coherence, and even values for the lead field matrix in beamformer techniques are predominantly a function of phase interference over aggregated samples rather than true sample-by-sample variations in source amplitude (*Alexander et al., 2013*; *Alexander et al., 2015*). Conversely, if the amplitude recorded at contacts is sufficiently small, the estimated phase values will not affect the estimates of SF produced here; they will be noise-like and not contribute to the dominant components of the SVD. We introduce the effects of TF amplitude as an addendum to our main analyses.

## SVD for empirical fourier decomposition

Fourier methods are commonly used in scientific research to analyze signals that extend in time and space. The methods use sinusoids as a basis to decompose arbitrary n-dimensional series into weighted sums of sinusoids. Fourier methods have been used to analyze TWs in the EEG (*Alamia et al., 2023*; *Alamia and VanRullen, 2019*). The non-uniform spacing of the measurement array in the standardized montage (e.g. 10–20 system) is overcome by interpolating a regular grid between measurement sites. In sEEG, however, the contact spacing is irregular to a much larger degree, making interpolation not an option. Fourier analysis with multi-resolution grids is still an active area of research (*Tong et al., 2003*; *Gaspar et al., 2009*), yet these approaches are limited to a few scales of grid and do not account for irregular sampling. The analyses provided by nonuniform discrete Fourier transform methods (*Margrave and Ferguson, 1997*; *Candes et al., 2008*; *Ying, 2009*; *Ruiz-Antolín and Townsend, 2018*) are also not suited to the extremes of sparsity and irregularity of the gray matter sEEG contacts analyzed here.

SVD is a standard method to reorganize data into a more interpretable form. SVD has recently become a more common tool in neuroscience (*Pang et al., 2023*; *Sivakumar et al., 2016*; *Cabral et al., 2023*; *Daini et al., 2020*; *MacDowell and Buschman, 2020*; *Robinson et al., 2016*; *Rosso et al., 2025*); for an alternative view, see *Shinn, 2023*. Here, we used SVD primarily to discount the influence of phase estimates that do not contribute strongly to the spatial covariance of phase. Also, the LSVs can be interpreted as Fourier components with corrections for the irregularity of the measurement array.

We have previously shown that SVD can be used to decompose phase data obtained via EEG, MEG, or ECoG (*Alexander et al., 2006*; *Alexander et al., 2013*). The measurement coordinates of the scalp EEG or MEG sensor array define the (partial) surface of a three-dimensional spheroid. The SVD decomposes the phase data into families of phase gradients projected onto the surface of the spheroid. As illustrated in *Figure 3F*, the first three LSVs extracted using SVD are approximately equal in wavelength to the size of the measurement array. Since the LSVs were smooth gradients of phase, this meant these first three components were large-scale TWs. The first three components were (for EEG) anterior-posterior gradient, inferior-superior gradient, and left-right gradient (*Alexander et al., 2006*). In addition, macroscopic TWs with arbitrary directions could be described using weighted sums of the components, for example, a mixture of the anterior-posterior and left-right gradients gives a diagonally propagating macroscopic TW. The next three LSVs were approximately wave-number equal to two (i.e. twice the SF), and each successive three LSVs increasing in wavenumber by one (*Alexander et al., 2013*).

We can draw out the relation between SVD and Fourier analysis more explicitly. Consider phase data with a smooth spatial organization, where the peak in the SF spectrum is at long wavelengths (i.e. $\lambda \approx$ size of measurement array), and the spectral power monotonically decreases with increasing SF. A matrix of phase signals, **A**, then decomposes the many samples of spatial vectors of phase into a series of LSVs, $u_r$ in descending order of singular values, $\sigma_r$, considering only the $r$ highest singular values (*Cline and Dhillon, 2006*).

$$A = U_r \Sigma_r V_r^* = \sum_{i=1}^{r} \varsigma_i u_i v_i^* \tag{2}$$

The effectiveness of the SVD is that it reorganizes the data into components that are uncorrelated with each other (the LSVs), but that explain known amounts of variance in the data (the singular values). Because the correlation is zero, the components can be treated as independent observations of a system. Weighted sums of these components reproduce the original data in a lossless manner. Alternatively, only the most important components can be kept to more concisely describe the original data, while the ignored components can be treated as noise. One of the main reasons to use SVD in the present research is the reorganization of data into components that explain most of the variance, and others which are noise-like. This means the SVD discounts phase measurements that do not create consistent patterns of spatial covariation. This is a desirable effect, because being able to calculate phase at each time point and contact does not imply the phase is meaningful. By choosing a cut-off in the reduced-rank SVD model that excludes negligible sources of explanatory variance, the meaningless measurements of phase are excluded. This would not be the case if SF estimates were made directly on the phase estimates.

SVD has been described as a generalization of Fourier analysis (*Brunton and Kutz, 2022*), in that it is used to empirically recover a set of bases that allow the original data to be represented as weighted sums of that basis. We illustrate in *Figure 3* that analysis of suitable signals using SVD enables recovery of a set of bases that are approximations of sinusoidal basis functions (see also *Shinn, 2023* for a critical view). 'Suitable' here means (a) monotonically decreasing in power with frequency or calculated at a clear peak in the power spectrum, (b) containing no other strongly competing explanatory features (such as gradients) and, importantly, (c) smoothness of values in the underlying function. Suitable signals include digitized classical music, noise generated by random walk processes (proof in *Shinn, 2023*), and intra- and extracranial cortical measurements (*Alexander et al., 2019*; *Alexander et al., 2015*). This specific ability of SVD to recover approximately sinusoidal bases means that, though empirically driven, in the present research there is a close theoretical relation between SVD and Fourier analysis.

The reasons that SVD finds a sinusoidal basis are treated in *Shinn, 2023*, for the case of a uniform measurement array. Essentially, spatial smoothness in the signal means neighboring values quantify the curvature in the form of a central difference approximation of the second derivative. In the present data, the spatial smoothness is a function of real correlations in cortical activity across space, plus short-range volume conduction (<1 cm; *Kajikawa and Schroeder, 2011*; *Nelson et al., 2013*; *Dubey and Ray, 2019*; *Orczyk et al., 2021*). The continuous function corresponding to this difference approximation is the harmonic oscillator, which is satisfied by linear combinations of sines and cosines. Depending on the boundary conditions of the signal, the SVD components may also involve the first

order forward finite difference approximation of the first derivative. Here also, the corresponding function in the continuous case is also a sinusoid. Also, depending upon the boundary conditions, the LSVs may not be pure Fourier components but weighted sums of the terms in the Fourier series (*Shinn, 2023*).

In the present research, the vectors of phase (over the array) were decomposed into uncorrelated components using SVD. The matrix of complex-valued phases, $\Phi_{st}$, is submitted to the SVD. Two outputs from the SVD are used to compute the SF spectrum. These are the LSVs, which are each complex-valued vectors with indexes $s$. Each value within an LSV, $\zeta_s$, consisted of a phase, $\omega_s = \angle\zeta_s$, and a magnitude, $a_s = |\zeta_s|$, which could vary over the LSV, for example $a_s$ tend to be low at singularities (rapid rates of change) in $\omega_s$. The second output, the LSV scaling value, $\varsigma_r$, is obtained from the main-diagonal values of the scaling matrix $\Sigma_r$, and comprises the ordered contributions, or weightings, of each LSV's contribution to the original data.

Subsequent to the SVD, it is crucial to analyze the nature of the components, rather than just assume they are physically meaningful decompositions of the measurements (*Shinn, 2023*). This reality check arises the same sense for Fourier analysis, which is guaranteed to decompose any continuous signal into sums of sinusoids, even if the underlying physical process is not sinusoidal, such as in heat diffusion (*Fourier, 1822*). Here, we use the SVD as a preliminary step to then estimate the SF spectrum in sEEG, rather than make specific claims about the empirical meaningfulness of the components, although related methods and claims have been used in a wide range of neuroscience research (*Pang et al., 2023*; *Sivakumar et al., 2016*; *Cabral et al., 2023*; *Daini et al., 2020*; *MacDowell and Buschman, 2020*; *Robinson et al., 2016*).

The threshold for exclusion of 'noise' components is justified by the wide range of dimension reductions (between 10 and 40 components) that were trialed. The method does not depend on choosing a particular number of signal components. The noisier the component (from the point of view of spatially coherent patterns of phase), the lower its contribution to variance explained by the SVD (and hence to the SF calculation). In the results reported here, we used the first 14 LSVs and singular values.

Here, we describe in more detail the relation between Fourier analysis and SVD methods to introduce the logic of the method. We limit our description to the one-dimensional case, without loss of generality. For suitable signals, as defined earlier in this section, each of the first few LSVs, $v_k$, is an approximate complex sinusoid with wavenumber equal to $k$.

$$v_k \cong e^{-i2\pi k} \tag{3}$$

This means the SVD has decomposed the matrix of complex-valued phase into bases that approximate a Fourier decomposition in the spatial dimension

$$\psi_k = \sum_{n=0}^{N-1} \psi_n \cdot e^{-\frac{i2\pi}{N}kn} \tag{4}$$

where $\Sigma_n$ is the sum over the spatial array of length N samples. Note that this means the maximum SF that can be measured is equivalent in size to a one-cycle sinusoid (i.e. when $k=1$) over the sampling array.

Just as the LSVs play the same role as the last term in *Equation 4*, $e^{-\frac{i2\pi}{N}kn}$, so the singular values describe the weights of each sinusoidal term, corresponding to the sums over $\psi_n$ in Fourier analysis. The SF spectrum can therefore be estimated after the SVD by summing over the singular values assigned to unique (or by binning over a limited range of) SFs. This procedure is illustrated in *Figures 3 and 4*.

This correspondence, between Fourier analysis and SVD of spatially organized phase, allows SF spectra to be estimated on semi-regular arrays, such as EEG, MEG as well as ECoG arrays on the surface of the cortex (*Alexander et al., 2019*; *Alexander et al., 2016*; *Alexander et al., 2013*). The first dozen or so LSVs can be assigned a SF by calculation of the circular range of phase across the array. This assignment is possible because the maps of phase so-produced are smooth and regular, even if the measurement array is semi-regular (*Alexander et al., 2016*; *Alexander et al., 2013*). Note that the smooth maps of phase arise from the regularities (covariances) in the phase data itself and

do not require information about the spatial locations of the measurement sites. The measurement locations are only required, *post hoc*, to estimate the SF of the maps, in cycles/m.

We hypothesized from our previous observation regarding LSV of phase (*Alexander et al., 2006*; *Alexander et al., 2016*; *Alexander et al., 2013*) that the methods would generalize sufficiently well to estimate SF spectra in the case of highly irregular measurement arrays. *Ruiz-Antolín and Townsend, 2018* show that Fourier analysis can be performed on non-uniform arrays, provided each (irregularly placed) sampling site can be re-mapped to a sampling site on a uniform array. The key observation to their procedure is that the effects of the re-mapping can be well approximated as a low-rank matrix, and these effects can be computed by a limited number of additional, corrective, Fourier analyses. It seems reasonable then to assume that, just as SVD captures sinusoidal components on (quasi-) regular arrays, SVD can also capture components on highly irregular arrays that are combinations of sinusoidal and the (low-rank) effects of sampling site displacement. Such a mathematical proof is beyond the scope of the present research. Instead, we rely on surrogate testing to show that the present method accuracy assesses the shape of the SF spectra. We discuss this issue further at the end of the *Methods* section.

## Removal of DC component of phase spectrum

*Alexander et al., 2016* introduced a quantitative method for splitting spatio-temporal waves into positive and negative components. This enables the pure TW component to be measured, while removing the effects of standing wave activity, including global phase-locked synchrony. We apply this method to the LSVs of phase to account for the effects of average reference, which introduce an artifactual global synchrony, thereby inflating the lowest SF's power. Since we are considering the spatial spectra of phase, the removal of globally synchronous components has the effect of removing the DC component of the spectrum. In general, this component is outside the range of SFs of interest here (8–50 cycles/m), and inclusion or exclusion of this methodological step did not influence the present findings. It allowed us, nevertheless, to discount the DC component in the phase spectrum.

The positive SF component is calculated as follows. The procedure takes a vector of Fourier components (phase-only or including the amplitude) over the array. Specifically, the LSVs are used as input to this algorithm. The LSV, which has complex magnitude and phase, $r\phi_s$, is first extended to one full cycle of phase

$$r\phi_{sc} = r\phi_s \times \phi_c \tag{5}$$

where $\times$ is the Cartesian product, $\phi_c$ is a vector of unit length phases over $n$ samples, where

$$n = \lfloor f_s/f \rceil \tag{6}$$

and $f_s$ is the sampling frequency and $f$ is the center frequency of the phase estimate. We then perform an SVD on the real part of the temporally extended phase vector. The pure TW can then be estimated by combining the first and second LSVs into the real and imaginary parts of a complex-valued phase vector

$$\phi_p = e^{-i\angle(u_1 + iu_2)} \tag{7}$$

where $u$ are the LSVs of R ($r\phi_{sc}$). Examples of the LSVs of $\Phi_{st}$ for the MEG data, before and after the removal of standing wave effects, are shown in *Figure 6—figure supplement 4*.

## Multi-scale differencing of phase

For the sEEG, the extreme irregularity of the array means the smoothness of the phase in the LSVs of $\Phi_{st}$ breaks down. To overcome this limit, we introduce further numerical analyses based on multi-scale differencing. We used multi-scale difference methods since we were interested in estimating the entire SF spectrum, including global ranges. The accuracy of the multi-scale differencing method was compared to SF spectra obtained using an alternative measurement modality (MEG) and using surrogate data whose ground truth was known by construction. Surrogate testing indicated that use of nearest neighbor differences alone was not sensitive to lower SFs known to be present in the data. Further, we demonstrate an important effect of the distortions introduced by the irregularity of sEEG contacts to smooth phase maps calculated via SVD. These distortions meant the largest inter-contact

distances were most sensitive to low SF parts of the signal and the smallest contact distances were most sensitive to high SFs. This observation differs from what is found using semi-regular arrays (*Alexander et al., 2006*; *Alexander et al., 2013*), such as those obtained from MEG (*Figure 3F*). Regular or semi-regular measurement arrays provide the same estimates of rate of change of phase per meter at all spatial scales of differencing, because of their uniform rates of change in phase with distance.

The SF contributions of each LSV were quantified by considering triplets of phase values, in the form of approximately equilateral triangles mapped onto the folded cortical surface. The rate of change of phase per meter, across triangles of contact in each LSV, was used for each estimate of SF. The results therefore did not directly require the inference of spatial oscillatory components.

We group contacts into approximately equilateral triangles so the direction of phase flow could be unambiguously determined. Comparing phase between pairs of contacts, such as in phase coherence (*Srinivasan et al., 1998*; *Bullock et al., 1995*), along a line of contacts (*Alamia et al., 2023*; *Burkitt et al., 2000*; *Ito et al., 2005*), or for a highly elongated triangle, only measures the phase flow along the axis in which the contacts are aligned. This procedure can greatly overestimate the rate of change of phase, in the case where the flow has a strong orthogonal component, that is, directed across the axis on which the contacts are arranged. We therefore reorganize the irregular field of sensors into triplets of approximately equilateral triangles.

We collated the unique triangles, that is contacts could appear in multiple triangles. First, the contact's position was projected to the nearest point on the surface of the cortex. We then computed the geodesic distance (*Margulies et al., 2016a*) between every pair of contacts using the surface mesh derived from the participant's MRI scan. Triangles were counted as approximately equilateral if the most acute angle was greater than $\pi/4$. Calculating geodesic distances enabled the true distances along the surface of the cortex to be accounted for, and, since the TWs move through the cortical sheet, this is presumed more accurate than taking the Cartesian distances from the folded cortex. Use of the geodesic distance measure—which is only defined within a hemisphere—also meant the triangles were only defined within hemispheres, that is a triplet of contacts could not span both hemispheres. Contacts in the sEEG participants often exist in both hemispheres. However, the SVD step allowed patterns of phase to be detected from the regularities (covariances) across hemispheres. Examples of geodesically defined triangles are shown in *Figure 5*, *Figure 5—figure supplement 1*, from both large and small size triangle bins.

The number of triangles varied from 11051 down to 66 across participants, resampled from a range of gray matter contacts 108 down to 31. Triangle SF, $\xi_T$, is given by one over the triangle linear size (square-root of the area of the triangle):

$$\xi_T = \frac{1}{\sqrt{|\bar{AB} \times \bar{CA}|/2}} \tag{8}$$

$\bar{AB}$ and $\bar{CA}$ are edge vectors of the triangle with vertices *A*, *B*, *C* and × is the cross-product. *A* is nominally at coordinate (0,0) within the triangle, and *B* and *C* are computed via the relevant geodesic distances across the cortical sheet. The SF distribution of triangle sizes formed a peaked distribution with maximum triangle counts occurring near the large end of triangle size range (see *Figure 5—figure supplement 1*). An effective uniform distribution of $\xi_T$ was achieved by dividing the range of $\xi_T$ into 32 equally spaced SFs, corresponding to triangle sizes from 32 cm to 1 cm. An additional bin with the smallest triangle sizes was allowed to include the rare triangles of size <1 cm and, depending on the participant, the lowest $\xi_T$ bins could be empty if the contacts spanned a more limited range than 16+cm. A normalization term, $N_T$ was then applied, such that the weight of each triangle's contribution to the SF calculation was inversely proportional to the number of triangles in that triangle's bin. This meant the SF estimates of phase were undertaken using an underlying set of distances with an effective uniform distribution in $\xi_T$.

For each LSV, and for each triangle of three contacts, we computed the rate of change in phase per meter on the triangle in the abstracted plane implied by the three inter-contact geodesic distances. Hence the SF, $\xi$, is given by:

$$\xi = \frac{|\omega \nabla_H|}{2\pi} \tag{9}$$

where $\nabla_H$ is the horizontal (i.e. two-dimensional) del-operator, so

$$\omega\nabla_H = \begin{bmatrix} \dfrac{\partial\omega}{\partial x} \\[2ex] \dfrac{\partial\omega}{\partial y} \end{bmatrix} \tag{10}$$

and $\omega=\angle\zeta$. Each vector in *Equation 9 and 10* has an implicit triangle index, $T$, which indexes a triple of contacts drawn from $s$, and LSV index, $r$, so $\xi$ can be rewritten as $\xi_{rT}$.

The weighting of each triangle's contribution to the SF spectrum is computed as a function of the scaling value of the $r^{th}$ LSV (i.e. the amount of variance explained), the magnitude of the LSV's complex value at the triple of relevant contacts and the normalization term for each triangle, and is given by:

$$W_{rT} = \varsigma_r \langle a \rangle_T N_T \tag{11}$$

where $\langle a \rangle_T$ is the mean element-wise magnitude of $\zeta_s$ for the subset of three vertices in $s$ for LSV $r$. Combining the SF estimate for each geodesic triangle in each LSV (*Equation 9*) with the weighting of each triangle's contribution to the power (*Equation 11*), allows the construction of the estimated SF spectrum. For $K$ SF bins, we have a bin width, $w$:

$$w = \left\lceil \frac{max\,(\xi) - min\,(\xi)}{K} \right\rceil \tag{12}$$

with each $i^{th}$ weight $W_{rT}$ being assigned to the bin corresponding to its computed $i^{th}$ SF, $\xi_{rT}$. The estimated power in arbitrary units (A.U.) for the $k^{th}$ bin, $k \in [1, K]$ is then,

$$P_k = \sum_i W_i;\, i \in \left\{ j \,\middle|\, kw \leq \xi_i\, (k+1)w \right\} \tag{13}$$

We used 32 SF bins to plot the SF spectra, after choosing $min\,(\xi)$ and $max\,(\xi)$ on a reasonable range (e.g. SFs equivalent to twice the maximum triangle size through to 50 cycles/m for plotting the sEEG data; SFs equivalent to the maximum triangle range through to 50 cycles/m for computing regression lines). For some additional analyses, for example the effects of average referencing on the spectrum, the lowest SF plotted was 0.5 cycles/m. The estimated power is in arbitrary units because it is a function of the number of contacts submitted to the SVD and number of components retained. Since these are both constant for the calculation of a given spectra, the relative power and different SFs are the relevant quantity for each participant.

## Surrogate testing

The primary method of surrogate testing introduced an artificial SF component to the sEEG signal, whose ground truth was known. The artificial SF components were created by inflating the cortical mesh of each participant to a sphere, then projecting a phase gradient onto the sphere, in either the anterior-posterior or left-right directions. The phase gradients were varied in wavelength from 2 to 16 cycles/m, and the positions of the actual contacts projected onto the inflated sphere used to infer surrogate basis vectors. Surrogate right singular vectors (i.e. time-series of weightings for the LSVs) were constructed by reversing the time-order of the empirical right singular vectors for that participant at the relevant TF, then taking the complex conjugate to re-establish the correct direction of phase change over time. A surrogate time series in complex-valued phase was constructed using each surrogate basis vector as LSV, a scaling value of 1.0 and a surrogate right singular vector, according to *Equation 2*. The surrogate time series in phase was added to the empirical phase signal at each TF, with the surrogate signal weighted, as $w_{sur}$, across a range of values to both (1) perturb the empirical signal (1/3 surrogate to 2/3 signal ratio), and (2) to see the effects of perturbation by realistic signals on the veridical surrogate phases (7/8 surrogate to 1/8 signal ratio).

$$\Phi_{sur} = e^{-i\angle\left(\Phi + w_{sur}\Phi_\xi\right)} \tag{14}$$

where $\Phi_\xi$ is the phase surrogate signal at some SF and direction and $\Phi_{sur}$ is the combined signal.

The maximum range of volume conduction in local field potential measurements is less than 1 cm (*Kajikawa and Schroeder, 2011*; *Nelson et al., 2013*; *Dubey and Ray, 2019*; *Orczyk et al., 2021*). The maximum SF assessed in this study was 50 cycles/m, corresponding to a cortical distance of 2 cm for one cycle. For this reason, we did not add a further local blurring to the surrogate signal to mimic volume conduction in the gray matter. The effects of volume conduction on local field potentials are outside the detection range of the present methods and would have negligible effect on the surrogate signals applied.

We additionally analyzed two participants from a different MEG data set (*Zvyagintsev et al., 2008*), as a source of neurophysiologically realistic time series with known SF spectra (*Alexander et al., 2016*). Surrogate testing of the method using MEG was included because the large number of quasi-regularly spaced contacts enables recovery of smooth wave maps using SVD, with visually apparent SF values for each map. We then used the present methods to estimate the distinct frequency spectra for each individual LSV. In addition, we constructed MEG surrogates using the first 28 LSVs of phase, but reversing the relative weighting of the singular values, that is $\varsigma_i$, in *Equation 2*. This produced an MEG surrogate signal, in which small spatial scales of activity dominated the signal, giving a *k spectrum* (as opposed to the usual 1 /*k* spectrum, where *k* is the wavenumber). We additionally made use of the large number of contacts in the MEG array to test the effects of sparsity of measurement contacts. The phase data for the *k* spectrum surrogate, along with the unmodified MEG phase data, were further evaluated by randomly removing subsets of contacts until only 1/3 remained (51 of 151). This analysis took advantage of the large number of sensors available with MEG, so that highly irregular arrays are constructed by increasing the sparsity. This analysis was to rule out that the estimated SF spectra were artefactually created by the sparsity of the sEEG arrays, when analyzed using our method. We also used these MEG time series to test the effects of an average reference on our method, since with MEG the case of no reference is also available to analyze.

## Effects of TF power

The main analyses here were undertaken using complex-valued phase but generalize naturally to complex values that are not unit length, for example by not excluding TF power. Each value computed by the Morlet wavelet is denoted $r\phi=re^{-i\theta}$ where $r$ is the amplitude of the signal. The matrix $R\Phi_{st}$ was submitted to the SVD, and the SF spectra were estimated. There is inherent noise in the TF power, mostly due to the accuracy and residual uncertainty of contact placement relative to the *mm* scales of the cortical sheet. This results in large variation in the mean amplitude signal across contacts. For this reason, we also computed the normalized amplitude at each contact, $\bar{r}$, such that the normalized amplitude at a given contact varied between zero and one over the recording, and $\bar{R}\Phi_{st}$ was submitted to the SVD.

## Statistical analysis

Statistical testing of linear regressions reported in the results was undertaken as correlation between the empirical values ($y$) and the regression model values ($\hat{y}$). For regressions of SF versus estimated power, linear fits of the model, $\rho$, were undertaken using a log-log scaling of wavelength (1.0/SF) with power. This combination produced the best linear fits. The statistical significance of these correlations, over multiple comparisons, was tested by permutation methods. Each permutation of the data comprised a correlation between ys in the regression and random shuffling of $\hat{y}s$ (10,000 permutations). The probability of arriving at a mean correlation (ignoring the sign of the mean), $\langle\rho\rangle_n$, over $n$ frequencies, participants, or other analysis dimension was compared to the actual (unsigned magnitude of the) mean correlation (i.e. a two-tailed test). The tests for the significance of a family of regressions are reported as 'variable A versus variable B (permutation test over n-condition, p-level)'. The p-level was set to p<0.05 for planned analyses (individual participant regressions of wavelength versus power, over all TFs) and p<0.001 for others. When $n$ was one, that is only one correlation rather than the mean of a group of correlations, this is indicated by omission of the n-condition term.

## Effects of triangle size on SF estimates

We have assumed that the LSV derived from irregular arrays can be understood as Fourier components along with low rank 'corrections' for contact displacements (*Ruiz-Antolín and Townsend, 2018*). As part of our methodological approach to apply multi-scale difference estimates of SF, we examined

peak SF as a function of triangle size. Examination of SF spectra revealed that triangle bin size and SF peaks derived from each bin had a strong linear relationship. That is, small triangles were more sensitive to high SFs, large triangles were more sensitive to low SFs. Statistical analysis of spectra produced from each separate triangle size bin showed that triangle size (as mean triangle size over a bin) linearly increased with the peak in the SF spectra for that bin size (as logarithm of wavelength; permutation test over participants, p<0.001). Examples of this effect are shown in *Figure 3—figure supplement 1*. This effect is not expected under standard Fourier analysis on regular grids, for which the SF estimates will not vary if calculated by difference methods at alternative values for delta-distance. Likewise, the SF estimates from standard Fourier analysis will not substantively differ at most (except highest) SFs if the data are sampled with different underlying but uniform contact spacings. This effect arises in the present sEEG data due to the irregular spacing of the contacts, which become encoded in the LSVs of the SVD.

We analyzed this effect with surrogate methods, manipulating the inputs to the SVD of the random walk surrogate shown in *Figure 2A*. We performed the same SVD while systematically changing the distribution of intercontact distances in the measurement array. The results are shown in *Figure 3—figure supplement 2*. A sparse, uniform array, covering the same spatial range as our first analysis, retrieved the same SF components (panel **F**, column 1). A dense array with a narrow range of contacts retrieved only high SF components (panel **F**, column 5), relatively speaking, but exactly the components defined by the maximum array size. These results are expected from considerations from standard Fourier analysis. Intermediate cases of grid spacing were produced via modulo rearrangements of the contacts (*Sheridan et al., 2000*). The maximum contact range and the Nyquist frequency were varied systematically between the two extremes. Careful visual inspection indicates that SVD produced LSVs with high SF components that were a function of the minimum intercontact distance; this is shown by the rate of change of values in the LSVs in runs of nearest neighbor contacts (panel *F*, columns 2–4). However, lower SF representations were also present in the extracted components; this is demonstrated by interpolating between equivalent points in each run of nearest neighbor contacts (panel *G*).

Consider the application of a simplified, one-dimensional version of the present methods, that is estimating SF spectra from the LSVs via multi-scale contact distances rather than triangle sizes. When using a multi-scale contact array, the highest SFs can be extracted from smallest neighboring intercontact distances. The lowest SFs can be extracted from the largest intercontact distances. This effect arises from the way the SVD reorganizes the Fourier information under conditions of non-uniform sampling (cf. *Ruiz-Antolín and Townsend, 2018*). This is the same effect we see for triangle size in the sEEG data, that is small triangles are sensitive to high SFs and large triangles to low. We therefore used multi-scale differencing to estimate the SF spectra, in order to capture the full range of SF information present in each LSV. The accuracy of this approach was then confirmed by the surrogate testing of the methods.

## Acknowledgements

This project has received funding from the European Research Council (ERC) under the European Union's Horizon 2020 research and innovation programme (grant agreement No 852139 – Laura Dugué). We want to thank to Dr. R Austin Benn for helpful advice on measuring geodesic distances in the cortex, and to Dr. Kirsten Petras for reading versions of the manuscript and providing useful feedback.

## Additional information

### Funding

| Funder | Grant reference number | Author |
|---|---|---|
| European Research Council | European Union's Horizon 2020 research and innovation programme (grant No 852139 - Laura Dugué) | Laura Dugué |

The funders had no role in study design, data collection and interpretation, or the decision to submit the work for publication.

### Author contributions

David M Alexander, Conceptualization, Data curation, Software, Formal analysis, Validation, Investigation, Visualization, Methodology, Writing – original draft, Project administration; Laura Dugué, Conceptualization, Resources, Supervision, Funding acquisition, Validation, Investigation, Writing – original draft, Project administration, Writing – review and editing

### Author ORCIDs

David M Alexander ⓘ https://orcid.org/0000-0003-4583-8950
Laura Dugué ⓘ https://orcid.org/0000-0003-3085-1458

### Ethics

The study uses data from the RAM public database: https://memory.psych.upenn.edu/RAM_Public_DataInformed consent and ethic approval information can be found in: Ezzyat Y, Kragel JE, Burke JF, Levy DF, Lyalenko A, Wanda P, et al. Direct Brain Stimulation Modulates Encoding States and Memory Performance in Humans. Curr Biol. 2017 May 8;27(9):1251-8.

Reviewer #1 (Public review): https://doi.org/10.7554/eLife.100674.4.sa1
Reviewer #3 (Public review): https://doi.org/10.7554/eLife.100674.4.sa2
Author response https://doi.org/10.7554/eLife.100674.4.sa3

---

## Additional files

### Supplementary files

MDAR checklist

### Data availability

The key Python routines used in this research can be found here: https://github.com/DugueLab/Traveling-wave-analysis/blob/main/estimation_of_SF_of_phase_of_cortical_activity_on_irregular_measurement_arrays.ipynb (copy archived at *Alexander, 2026*).

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
