## [Editor Report · eLife Assessment]

This study introduces a novel method for estimating spatial spectra from irregularly sampled intracranial EEG data, revealing cortical activity across all spatial frequencies, which supports the global and integrated nature of cortical dynamics. It showcases **important** technical innovations and rigorous analyses, including tests to rule out potential confounds. However, further direct evaluation of the model, for example by using simulated cortical activity with a known spatial spectrum (e.g., an iEEG volume-conductor model that describes the mapping from cortical current source density to iEEG signals, and that incorporates the reference electrodes and the particular montage used), would even further strengthen the **solid** evidence.

---

## [Referee Report · Reviewer #1 (Public review)]

Summary:

The paper uses rigorous methods to determine phase dynamics from human cortical stereotactic EEGs. It finds that the power of the phase is higher at the lowest spatial phase. The application to data illustrates the solidity of the method and their potential for discovery.

Comments on revisions:

The authors have provided responses to the previous recommendations. The paper does not seem to contain further significant improvements. I am thus not inclined to change my judgement.

---

## [Referee Report · Reviewer #3 (Public review)]

Summary:

The authors propose a method for estimating the spatial power spectrum of cortical activity from irregularly sampled data and apply it to iEEG data from human patients during a delayed free recall task. The main findings are that the spatial spectra of cortical activity peak at low spatial frequencies and decrease with increasing spatial frequency. This is observed over a broad range of temporal frequencies (2-100 Hz).

Strengths:

A strength of the study is the type of data that is used. As pointed out by the authors, spatial spectra of cortical activity are difficult to estimate from non-invasive measurements (EEG and MEG) and from commonly used intracranial measurements (i.e. electrocorticography or Utah arrays) due to their limited spatial extent. In contrast, iEEG measurements are easier to interpret than EEG/MEG measurements and typically have larger spatial coverage than Utah arrays. However, iEEG is irregularly sampled within the three-dimensional brain volume and this poses a methodological problem that the proposed method aims to address.

Weaknesses:

Although the proposed method is evaluated in several indirect ways, a direct evaluation is lacking. This would entail simulating cortical current source density (CSD) with known spatial spectrum and using a realistic iEEG volume-conductor model to generate iEEG signals.

Comments on revisions:

I would like to clarify two points:

(1) In their response, the authors frame the role of simulations primarily as a means of assessing the effects of volume conduction. However, the purpose of evaluating a proposed estimation method through simulations extends beyond this specific issue. More generally, simulations are essential for establishing that the proposed method-particularly given the multiple non-trivial transformations applied to the observed data-produces accurate and reliable estimates under controlled conditions.

(2) The authors seem to interpret my use of the term current source density as referring to the current source density (CSD) method, which is an approach to mitigating volume conduction by inverting Poisson's equation. This was not my intention: current source density refers to the physical quantity (i.e., the spatial density of current sources) underlying macroscopic brain activity, and is independent of any specific estimation or inversion technique.

---

## [Author Response]

The following is the authors’ response to the previous reviews

**Public Reviews:**

**Reviewer #1 (Public review):**
Summary:The paper uses rigorous methods to determine phase dynamics from human cortical stereotactic EEGs. It finds that the power of the phase is higher at the lowest spatial phase. The application to data illustrates the solidity of the method and their potential for discovery.Comments on revised submission:The authors have provided responses to the previous recommendations.

We thank the reviewer for reviewing our manuscript again, and for their positive evaluation.

**Reviewer #3 (Public review):**
Summary:The authors propose a method for estimating the spatial power spectrum of cortical activity from irregularly sampled data and apply it to iEEG data from human patients during a delayed free recall task. The main findings are that the spatial spectra of cortical activity peak at low spatial frequencies and decrease with increasing spatial frequency. This is observed over a broad range of temporal frequencies (2-100 Hz).Strenghs:A strength of the study is the type of data that is used. As pointed out by the authors, spatial spectra of cortical activity are difficult to estimate from non-invasive measurements (EEG and MEG) and from commonly used intracranial measurements (i.e. electrocorticography or Utah arrays) due to their limited spatial extent. In contrast, iEEG measurements are easier to interpret than EEG/MEG measurements and typically have larger spatial coverage than Utah arrays. However, iEEG is irregularly sampled within the three-dimensional brain volume and this poses a methodological problem that the proposed method aims to address.Weaknesses:Although the proposed method is evaluated in several indirect ways, a direct evaluation is lacking. This would entail simulating cortical current source density (CSD) with known spatial spectrum and using a realistic iEEG volume-conductor model to generate iEEG signals.Comments on revised version:In my original review, I raised the following issue:"The proposed method of estimating wavelength from irregularly sampled three-dimensional iEEG data involves several steps (phase-extraction, singular value-decomposition, triangle definition, dimension reduction, etc.) and it is not at all clear that the concatenation of all these steps actually yields accurate estimates. Did the authors use more realistic simulations of cortical activity (i.e. on the convoluted cortical sheet) to verify that the method indeed yields accurate estimates of phase spectra?"And the authors' response was:"We now included detailed surrogate testing, in which varying combinations of sEEG phase data and veridical surrogate wavelengths are added together. See our reply from the public reviewer comments. We assess that real neurophysiological data (here, sEEG plus surrogate and MEG manipulated in various ways) is a more accurate way to address these issues. In our experience, large scale TWs appear spontaneously in realistic cortical simulations, and we now cite the relevant papers in the manuscript (line 53)."The point that I wanted to make is not that traveling waves appear in computational models of cortical activity, as the authors seem to think. My point was that the only direct way to evaluate the proposed method for estimating spatial spectra is to use simulated cortical activity with known spatial spectrum. In particular, with "realistic simulations" I refer to the iEEG volume-conductor model that describes the mapping from cortical current source density (CSD) to iEEG signals, and that incorporates the reference electrodes and the particular montage used.Although in the revised manuscript the authors have provided indirect evidence for the soundness of the proposed estimation method, the lack of a direct evaluation using realistic simulations with ground truth as described above makes that remain sceptical about the soundness of the method.

We thank the reviewer for reviewing our manuscript again.

We have reviewed the literature again on volume conduction effects in LFP measures of cortical activity. In all publications we reviewed, the conclusion is that the range of the effect is <1cm. We now mention the range of volume conduction in the Methods section dealing with the surrogate models (lines 1054-9) as well as added emphasis in the Discussion (lines 594-9).

The highest spatial frequency we consider in the present research is 50c/m, which corresponds to a cortical distance of 2cm. This is well outside the range of volume conduction effects in LFPs. Mathematically speaking, blurring (e.g. Gaussian) acts as a low-pass filter, attenuating higher spatial frequency components. But only for components within the spatial range of the Gaussian blurring i.e. for LFPs, higher than 100c/m. There will therefore be negligible effects (mathematically speaking, zero effect) of volume conduction in the results reported by us. If the veracity of these studies on volume conduction with LFPs is accepted, then the reviewer’s requested simulation reduces to “estimating spatial spectra [using] simulated cortical activity with known spatial spectrum.” This is what we have done, in a direct and simple manner.

If the ubiquity and importance of spatio-temporal dynamics in cortex is accepted, then it is insufficient to describe “the mapping from cortical current source density (CSD) to iEEG signals”, since this presumes a model of cortical activity that does not capture the correlations in space and time that we assume are critical to cortical function. We are aware the CSD approach has a long and successful history of unravelling brain mechanisms. However, an emphasis on traveling waves (and spatio-temporal dynamics in general) is in part a challenge to this approach (and the idea of localized sources in general). CSD approaches carry similar assumptions (but at a smaller scale, <1cm) as those elaborated in Zhigalov and Jensen (2023) for extra-cranial measures. In both cases, removal of volume conduction effects emphasizes standing wave activity (localized static, oscillatory sources) over traveling wave activity. In this manner, these methods tend to confirm their starting assumptions (as does our own approach, of course). What is required is external empirical validation to break any circular confirmation of initial theoretical choice of basis. All this is a way of saying that CSD approaches are not the unproblematic, direct methods that the reviewer asserts.

We did understand the reviewer’s request to model the effects of volume conduction. Our own view of realistic cortical simulations differs from the reviewer’s, setting aside the final step in the forward modeling pipeline which would add the effects of volume conduction in the grey matter. By simulating real-time dynamics, it should be possible to untangle the effects of volume conduction from true spatio-temporal correlations. This is because the volume conduction effects are essentially instantaneous, compared to the relatively slow motion of traveling waves. So, the measurement of purely spatial phase vectors is prone to smearing artefact, but following the trajectory of a wave over one cycle can more accurately determine the range of true interactions. One could, for example, compare the usual CSD forward modelling with TWs in simulations, see which is the best predictor of future activity, and compare these to empirical measurements. Here, the CSD analysis would remove the volume conduction effects but also emphasize standing activity over motion, even where the motion was veridical in the simulation.

Even so, these tests are only relevant in <1cm range.

Another issue is ephaptic coupling, which we mention in the discussion. This means that some of the local volume conduction effects are not merely artefacts from the point of view of cortical function, but have a real causal effect. The strength of the word ‘some’ has yet to be completely resolved in the literature, and it would be technically challenging to include these effects in any simulation.

Finally, simulation should be an adjunct to empirical studies, or used when empirical studies are not possible. We do not think, in this case, they are the ‘only direct’ way to evaluate our method. We, rather, rely on the converging evidence from empirical studies of volume conduction in LFPs which show this effect is outside the range of our reported results.